Forelimb muscle and joint actions in Archosauria: insights from Crocodylus johnstoni (Pseudosuchia) and Mussaurus patagonicus (Sauropodomorpha)

Otero Alejandro 1 2 alexandros.otero@gmail.com
Allen Vivian 3
http://orcid.org/0000-0002-9690-7517 Pol Diego 2 4
http://orcid.org/0000-0002-6767-7038 Hutchinson John R. 3
1 División Paleontología de Vertebrados, Museo de la Plata , La Plata, Buenos Aires , Argentina
2 Consejo Nacional de Investigaciones Científicas y Técnicas (CONICET) , Buenos Aires , Argentina
3 Department of Comparative Biomedical Sciences, Structure and Motion Laboratory, Royal Veterinary College , London , UK
4 Museo Egidio Feruglio , Trelew, Chubut , Argentina
Wilson Laura
Electronic publication date: 2017 Nov 24
Publication date: 2017
Volume: 5
Electronic Location ID: e3976
Received 2017 May 5; Accepted 2017 Oct 10
Copyright: © 2017 Otero et al.
Copyright year: 2017
Copyright holder: Otero et al.
License: This is an open access article distributed under the terms of the Creative Commons Attribution License, which permits unrestricted use, distribution, reproduction and adaptation in any medium and for any purpose provided that it is properly attributed. For attribution, the original author(s), title, publication source (PeerJ) and either DOI or URL of the article must be cited.
License URL: https://creativecommons.org/licenses/by/4.0/

Keywords: Dinosauria, Crocodylia, Musculoskeletal model, Quadrupedalism, Pronation, Range of motion, Posture, Bipedalism, Moment arm, Biomechanics

Funding: International Exchanges Program of the Royal Society European Research Council (ERC) 695517 Agencia Nacional de Promoción Científica y Tecnológica PICT 2015-0504 This research was possible through financial support provided by the International Exchanges Program of the Royal Society (to John R. Hutchinson and Alejandro Otero). This project has also received funding from the European Research Council (ERC) under the European Union’s Horizon 2020 research and innovation programme (grant agreement no. 695517) to John R. Hutchinson and Agencia Nacional de Promoción Científica y Tecnológica (PICT 2015-0504) to Alejandro Otero. The funders had no role in study design, data collection and analysis, decision to publish, or preparation of the manuscript.

==============================
Many of the major locomotor transitions during the evolution of Archosauria, the lineage including crocodiles and birds as well as extinct Dinosauria, were shifts from quadrupedalism to bipedalism (and vice versa). Those occurred within a continuum between more sprawling and erect modes of locomotion and involved drastic changes of limb anatomy and function in several lineages, including sauropodomorph dinosaurs. We present biomechanical computer models of two locomotor extremes within Archosauria in an analysis of joint ranges of motion and the moment arms of the major forelimb muscles in order to quantify biomechanical differences between more sprawling, pseudosuchian (represented the crocodile Crocodylus johnstoni) and more erect, dinosaurian (represented by the sauropodomorph Mussaurus patagonicus) modes of forelimb function. We compare these two locomotor extremes in terms of the reconstructed musculoskeletal anatomy, ranges of motion of the forelimb joints and the moment arm patterns of muscles across those ranges of joint motion. We reconstructed the three-dimensional paths of 30 muscles acting around the shoulder, elbow and wrist joints. We explicitly evaluate how forelimb joint mobility and muscle actions may have changed with postural and anatomical alterations from basal archosaurs to early sauropodomorphs. We thus evaluate in which ways forelimb posture was correlated with muscle leverage, and how such differences fit into a broader evolutionary context (i.e. transition from sprawling quadrupedalism to erect bipedalism and then shifting to graviportal quadrupedalism). Our analysis reveals major differences of muscle actions between the more sprawling and erect models at the shoulder joint. These differences are related not only to the articular surfaces but also to the orientation of the scapula, in which extension/flexion movements in Crocodylus (e.g. protraction of the humerus) correspond to elevation/depression in Mussaurus. Muscle action is highly influenced by limb posture, more so than morphology. Habitual quadrupedalism in Mussaurus is not supported by our analysis of joint range of motion, which indicates that glenohumeral protraction was severely restricted. Additionally, some active pronation of the manus may have been possible in Mussaurus, allowing semi-pronation by a rearranging of the whole antebrachium (not the radius against the ulna, as previously thought) via long-axis rotation at the elbow joint. However, the muscles acting around this joint to actively pronate it may have been too weak to drive or maintain such orientations as opposed to a neutral position in between pronation and supination. Regardless, the origin of quadrupedalism in Sauropoda is not only linked to manus pronation but also to multiple shifts of forelimb morphology, allowing greater flexion movements of the glenohumeral joint and a more columnar forelimb posture.

Introduction

Archosauria (all descendants of the most recent common ancestor of Crocodylia and Aves) has been a highly diverse and disparate clade since the Triassic period (<250 Ma), both morphologically and ecologically. This diversity and disparity is reflected not only in the great abundance and taxonomic richness that Archosauria achieved in the past, but also in its living representatives. Terrestrial locomotion in extant archosaurs (crocodiles and birds) is split between two extremes—‘sprawling’ (less erect; Gatesy, 1991) quadrupeds and parasagittally erect bipeds (not to mention amphibious habits vs. flight). The evolutionary patterns that preceded and gave rise to these disparities have long been an attractive research subject (Romer, 1956; Jenkins, 1993; Gatesy & Middleton, 1997; Hutchinson & Allen, 2009; Gauthier et al., 2011; Bates & Schachner, 2012), including the study of topics such as the transition from bipedalism to quadrupedalism (Bonnan & Yates, 2007; Maidment & Barrett, 2011), and the origin of avian flight (Jenkins, 1993; Dial, 2003), among others.

The functional anatomy of these locomotor transitions has also attracted considerable research effort (Romer, 1956; Jenkins, 1993; Gatesy & Middleton, 1997; Hutchinson & Allen, 2009; Gauthier et al., 2011; Bates & Schachner, 2012). Much of the attention has focused on the evolution of the hindlimb in Dinosauriformes as it adapted to the demands of bipedal locomotion (Romer, 1923; Carrano, 2000; Hutchinson & Gatesy, 2000; Hutchinson et al., 2005; Bates & Schachner, 2012), particularly in the theropod lineage (Gatesy, 1990). However, archosaur forelimbs have also undergone major functional transformations in Archosauria. Fewer studies have dealt with changes in forelimb function during the quadruped (e.g. basal archosaurs) to biped (e.g. basal sauropodomorphs) transition (but see Hutson & Hutson, 2013, 2014, 2017; Hutson, 2015). The lineage of Triassic archosaurs leading to sauropods began as quadrupeds, transitioned to bipedality close to the base of Dinosauria, and then shifted back to quadrupedality close to or at the base of Sauropoda (Wilson & Sereno, 1998; Carrano, 2005). The evolution of bipedalism itself has been a rare event, and such reversion to quadrupedalism from bipedalism is extremely rare, with known examples confined exclusively to the Dinosauria: sauropods themselves, and independently in three branches of ornithischian dinosaurs (ceratopsians, ornithopods and thyreophorans (Carrano, 1998; Brusatte et al., 2010; Maidment & Barrett, 2012; VanBuren & Bonnan, 2013; Hutson, 2015)).

Along the archosaur lineage leading to sauropods, forelimbs thus evolved from a role as weight-bearing locomotor modules to a variety of grasping and manipulating functions, before re-evolving weight-bearing and locomotor capacity with the transition back to quadrupedalism (Cooper, 1981; Bonnan & Senter, 2007; Bonnan & Yates, 2007; Yates et al., 2010; VanBuren & Bonnan, 2013). The biped–quadruped transition occurred between basal sauropodomorphs and basal sauropods, near the boundary of the Triassic and Jurassic periods (ca. 200 Ma). Consequently, the forelimbs of basal sauropodomorphs have captured the attention of palaeontologists because their functional morphology was likely pivotal to the acquisition of quadrupedalism (Bonnan & Yates, 2007). Previous studies of the anatomical and functional evolution of archosaur forelimbs have focused on reconstructing their general role in locomotion (Ostrom, 1974; Cooper, 1981; Johnson & Ostrom, 1995; Dodson & Farlow, 1997; Paul & Christiansen, 2000; Schwarz, Frey & Meyer, 2007; Bonnan & Senter, 2007; Bonnan & Yates, 2007; Maidment & Barrett, 2012; Fujiwara & Hutchinson, 2012; Baier & Gatesy, 2013). Recently, studies have begun to focus on the evolution of manual pronation across the biped–quadruped transition (Bonnan & Yates, 2007; VanBuren & Bonnan, 2013; Hutson, 2015). A more fully pronated manus (i.e. the palm facing ventrally/caudally) was hypothesized to be necessary to effectively produce braking or propulsive forces at the manus–ground interface (Bonnan & Yates, 2007).

Considering that the ancestral condition of the manus in bipedal dinosaurs appears to have been more supinated, with palms that faced medially (i.e. ∼90° from fully supinated) rather than caudally, the evolution of a fully pronated manus is thought to have been crucial to the origin of quadrupedalism in both sauropods and ornithischians (see references above). In particular, the degree to which the the morphology of the ulna and radius (antebrachium; i.e. forearm) would have permitted pronation in basal sauropodomorphs, and if so how widespread this ability was across the group, remain crucial questions in understanding the evolution of sauropod locomotion. However, the timing and sequence of changes in the functional anatomy of the forelimbs that were involved in the evolution of sauropod locomotion remain unclear, partly because to date the biomechanical factors involved have largely been analysed using only qualitative, two-dimensional methods (but see Reiss & Mallison, 2014).

Here, we use three-dimensional biomechanical computer models to quantitatively analyse the evolution of forelimb anatomy and function from early archosaurs to later sauropodomorphs. We model an adult Mussaurus patagonicus Bonaparte & Vince, 1979, a well-preserved representative basal sauropodiform (sensu Sereno, 2007), close to the origin of Sauropoda (Otero & Pol, 2013; Otero et al., 2015), and the extant Australian freshwater crocodile, Crocodylus johnstoni Krefft, 1873, a long-tailed quadruped reasonably representative of the ancestral archosaurian condition (Parrish, 1986; Gatesy, 1990). The phylogenetic relationship between Crocodylus and Mussaurus is presented in Fig. 1.

Figure 1 Simplified cladogram of crown group Archosauria depicting the relationships between Crocodylus johnstoni and Mussaurus patagonicus.

Modified from Brusatte et al. (2010) and Otero et al. (2015).

As in prior studies of archosaurian hind limbs (Hutchinson et al., 2005, 2008; Bates, Benson & Falkingham, 2012; Bates & Schachner, 2012; Maidment & Barrett, 2012), we use these musculoskeletal models to analyse the relationship between joint angles and moment arms for the muscles of the limb (An et al., 1984; Murray, Delp & Buchanan, 1995; Holzbaur, Murray & Delp, 2005), as well as possible joint ranges of motion (Reiss & Mallison, 2014). By quantifying similarities and differences in estimated limb biomechanical properties from our Crocodylus and Mussaurus models, we can explicitly evaluate how forelimb muscle actions and joint mobility may have changed with posture from basal archosaurs to early sauropodomorphs. ‘Action’ here is used as a shorthand term for moment arms about particular joints; distinguished from ‘function’ which would ideally involve broader data such as muscle force output, length change, etc. (Zajac, 1989; Allen et al., 2014). Our analysis considers these key questions: (1) How did forelimb musculoskeletal anatomy evolve between early, quadrupedal archosaurs (approximated by Crocodylus) and early sauropodomorphs such as Mussaurus? (2) How did this alter muscle action and joint ranges of motion? Particularly, was forelimb pronation possible in early sauropodomorphs like Mussaurus? (3) What were the consequences of (1) and (2) for forelimb posture and function? Particularly, how many observed functional changes helped or hindered the use of the forelimbs in terrestrial locomotion? Might such changes relate to the transitions from sprawling quadrupedalism to erect bipedalism in dinosaurs, and/or to the subsequent evolution of graviportal quadrupedalism in sauropods?

Our study represents the first attempt at comparative analysis of three-dimensional forelimb joint ranges of motion and muscle moment arms between a quadrupedal and a (at least facultatively) bipedal archosaur. We synthesize our findings with a review and meta-analysis of research on the biped–quadruped transition in sauropodomorphs.

Materials and Methods

Digitization and musculoskeletal modelling

Our model-building procedure for Mussaurus and Crocodylus comprised five steps: (1) bone geometry acquisition, (2) joint axis estimation, (3) muscle reconstruction (Mussaurus only), (4) muscle path specification (using the results from the prior three steps), (5) joint range of motion (ROM) analysis, and (6) analysis of muscle moment arms (automatically calculated from the muscle paths by the modelling software (see Delp & Loan, 2000)).

Bone geometry acquisition

The remains of the basal sauropodomorph Mussaurus patagonicus comprise several specimens of different ontogenetic stages, from post-hatchlings to adults (Bonaparte & Vince, 1979; Otero, Pol & Powell, 2012; Otero & Pol, 2013). Our study here focused on the best-preserved and complete right forelimb of the adult specimen number MLP 68-II-27-1 (Museo de la Plata, La Plata, Argentina), which comprises the scapula, partial coracoid, humerus, ulna, radius, three distal carpal elements, five metacarpals, first and ungual phalanges of digit one, and first phalanx of digit two (Otero & Pol, 2013). A three-dimensional portable surface scanner (NextEngine®, Santa Monica, CA, USA) was used to digitize each bone of Mussaurus, obtaining a 3D bone file (.obj format); similar files were output from the CT scan data (see below) for Crocodylus. Meshlab software (Visual Computing Lab—ISTI—CNR, Pisa, Italy) was used to reduce the resolution of the original .obj files as needed. Each individual bone file was then imported to 3D Studio Max® software (Autodesk®, San Rafael, CA, USA) in order to articulate the shoulder girdle and forelimb and to define the degrees of freedom (DOF; i.e. the possible axes of mobility) of each joint. We obtained our Crocodylus johnstoni specimen from the St. Augustine Alligator Farm and Zoological Park (St. Augustine, FL, USA), where it had died of natural causes in captivity. This specimen was also used in studies by Allen, Paxton & Hutchinson (2009), Fujiwara & Hutchinson (2012) and Allen et al. (2014), and was approximately adult, with a total body mass of 20.19 kg. It was scanned using a Picker PQ 5000 CT scanner (axial 512 × 512 pixel slices at 2.5 mm thickness; 100 mA, 120 kVp, resolution 1.024 pixels mm−1) and segmented in Mimics software (Materialise, Inc., Leeuwen, Belgium) after CT scanning for simple 3D modelling in the aforementioned studies, especially Fujiwara & Hutchinson (2012), who reconstructed the major forelimb muscles in a computational model that we adapted here.

Joint axis estimation, reference pose and terminology

We first used the osteology of each bony joint to estimate the orientations of the 3D axes of that joint (Fig. 2). Those axes also set up the translations required to place the bones in relation to one another, from proximal to distal.

Figure 2 Crocodylus and Mussaurus models.

Joint axes for rotation (x, y, z) in the reference pose showing the whole forelimb (A–F) and manus (G–I) in cranial (A, B), dorsal (C, D, G), craniolateral (E, F), medial (H) and ventromedial (I) views. (A, C, E) are Crocodylus and (B, D, F–I) are Mussaurus. Joint axis ‘x’ (red) corresponds to pronation/supination; ‘y’ (green) corresponds to adduction/abduction; and ‘z’ (blue) corresponds to extension/flexion, based on the coordinate system described by Baier & Gatesy (2013).

Considering that some extinct archosaurs have rather simply shaped appendicular condylar areas, implying the presence of large amounts of epiphyseal cartilage (Fujiwara, Taru & Suzuki, 2010; Holliday et al., 2010; Bonnan et al., 2013), thickness of soft tissues between the joints needed to be accounted for. Consequently, we left 10% of the total forelimb length as free space for epiphyseal cartilage in Mussaurus, distributed between the three main limb joints (i.e. glenohumeral, elbow, wrist), following the estimates of Holliday et al. (2010).

Geometric objects were used to link adjacent segments, using spheres (gimbal/ball-and-socket) for the glenohumeral joints and cylinders (hinges) for the other joints. These objects established the centres of rotation of each joint, through which the axes of joint rotation were positioned. Next, we defined the rotational DOFs that were allowed around each joint axis. Although translation is known to occur in extant archosaur forelimb joints (namely the glenohumeral joint (Baier & Gatesy, 2013)), we judged the potential effects on moment arms to be relatively minor. For the glenohumeral and elbow joints, we inferred from the morphology that these joints might have three DOFs (extension/flexion, abduction/adduction, and pronation/supination) in both Crocodylus and Mussaurus. For Crocodylus, we also allowed three DOFs for the wrist joint, although only two DOFs for the wrist of Mussaurus, not allowing pronation/supination because the block-like configuration of the carpus (Galton & Upchurch, 2004) and the fixed radius against the ulna (VanBuren & Bonnan, 2013) probably precluded such motion. Finally, for the metacarpo-phalangeal (MCP) and interphalangeal (INP) joints (only for Mussaurus), we only allowed one DOF (extension/flexion), because the bony anatomy indicated that these joints acted almost exclusively as hinges. Our ROM analysis (below) then considered how large the potential angular excursions of these DOFs might have been.

In order to set the initial position of the models, a reference pose at which all joint angles were set at 0° was chosen (Hutchinson et al., 2005, 2015; Baier & Gatesy, 2013; Baier, Gatesy & Dial, 2013). Thus the reference pose constituted a starting point from which comparisons could be made, facilitating understanding of what any value for a joint angle represents (vs. this reference angle, a fully straightened limb orientation, with the forelimb extended laterally, perpendicular to the vertebral column and body’s craniocaudal axis) (Figs. 2A–2F).

The segments of the forelimb were positioned following Baier & Gatesy (2013), in which the humerus was laterally oriented, perpendicular to the vertebral column (0° flexion) and its long axis was parallel to the ground (0° abduction), whereas the axis connecting the medial and lateral distal condyles was parallel to the vertebral column and the deltopectoral crest pointed ventrally (0° pronation). The major (longitudinal) axis of the ulna and radius (antebrachium) was parallel to that of the humerus (0° extension/abduction), and again the mediolateral axes of the distal condyles were parallel to the vertebral column’s longitudinal axis. Unlike the model of Baier & Gatesy (2013), the curvature of the ulna was in a plane parallel to the long axis of the humerus (0° pronation/supination). Finally, the manus was oriented with the long axis of the metacarpus parallel to the long axis of the antebrachium (0° flexion and abduction), whereas the curvature of the ungual of the first manual digit was in the same plane as the long axis of the antebrachium (0° pronation) (Fig. 2).

Rotations away from 0° for each joint were defined as three successive rotations of the segment relative to the axis proximal to it (i.e. its reference position), in the order x (e.g. pronation/supination), y (e.g. abduction/adduction) and z (e.g. extension/flexion). Our models had right-handed coordinate systems, so pronation (around x), abduction (around y) and extension (around z) were negative values (i.e. of joint angle rotations), whereas supination, adduction and flexion were positive values.

Finally, the articulated forelimb model in the reference pose was exported to musculoskeletal modelling software (Software for Interactive Musculoskeletal Modeling [SIMM]; Musculographics, Inc., Chicago, IL, USA) (Delp & Loan, 1995, 2000), using custom MATLAB code (The Mathworks, Inc., Natick, MA, USA).

There is not a general consensus regarding anatomical terminology among tetrapods because of their great morphological disparity (Harris, 2004; Wilson, 2006). Caution is thus warranted when attempting to compare animals with sprawling (ancestral, at some level for Archosauria) vs. erect (derived) stances or postures (Gatesy, 1991; Padian, Li & Pchelnikova, 2010) because each one can imply a different typical orientation for homologous or corresponding bones. To partly circumvent this problem, Jasinoski, Russell & Currie (2006) used two terms for bone orientation: ‘developmental’ and ‘functional’ orientations. The term ‘developmental orientation’ refers to the ancestral (sprawling) state, which is equivalent to that often present in tetrapod embryos, especially forms with relatively plesiomorphic limbs (e.g. Crocodylus). The term ‘functional orientation’ corresponds to the typical, approximate standing positions present in adults, which vary in different groups of tetrapods according to their locomotor mode(s) used. Our model oriented the forelimb segments from the most proximal to distal ones, starting with the scapula. We thus chose a developmental position of the scapula, which means that the scapular blade was initially oriented vertically and the glenohumeral joint was caudolaterally oriented, as retained by extant crocodiles. Positioning Mussaurus’s scapula the same way as in Crocodylus ensured the same kind of movements (pronation/supination, extension/flexion, abduction/adduction) around the same axes in both models, in the starting configuration (i.e. reference pose).

As our reference (‘developmental’) pose did not necessarily reflect a biologically plausible pose (i.e. a pose that is mechanically allowed by their joints without risk of dislocation, or ‘functional’ orientation), a standardized, biologically plausible pose was also chosen in order to make realistic comparisons between taxa in terms of joint ranges of motion and moment arm analysis. Hence we used a ‘resting’ pose for both taxa, which was modified from the reference pose and represented an approximate in vivo plausible pose that was feasible for Crocodylus and (in our judgement based on the anatomy) Mussaurus. In our analysis, starting from the reference pose (all 0° values), the resting pose for Mussaurus was set at 5° of supination, 25° of adduction and −40° of extension for the glenohumeral joint; whereas 70° of flexion was chosen for the elbow. In Crocodylus (also starting from the reference pose), the same values were chosen as in Mussaurus, except for long-axis rotation (pronation/supination) at the glenohumeral joint, which remained at 0° (Fig. 3). These admittedly were subjective judgements based on the joint morphology and function, but were deemed far more plausible than the reference pose and thus more suitable for biological comparisons.

Figure 3 Muscle reconstruction.

Right forelimb musculoskeletal models for Mussaurus (A–C) and Crocodylus (D–F) models in the resting pose in lateral (A, D), medial (B, E), and dorsomedial (C, F) views. Scale bar: 10 cm.

Regarding the terminology for naming the DOFs in this study (Gatesy & Baier, 2005; Hutchinson et al., 2005; Baier & Gatesy, 2013), we used pronation and supination for long axis rotation, the former alluding to internal (medial) and the later to external (lateral) rotation. We expressed those DOFs relative to the axes of the reptilian saddle-shaped glenoid on which they were acting, no matter if the limb would be elevating, depressing, protracting or retracting (see also Baier & Gatesy, 2013; Baier, Gatesy & Dial, 2013). Hence the abduction/adduction axis lay parallel to the long axis of the glenoid and the extension/flexion axis was perpendicular to the long axis of the glenoid (Figs. 4A–4C).

Figure 4 Joint axis nomenclature for the glenohumeral joint used in this study.

Reference pose in Mussaurus patagonicus showing ‘X’ (A, B), ‘Y’ (C, D) and ‘Z’ (E, F) axes. Reference pose in Mussaurus showing the vertical scapular blade, with a caudal orientation of the glenohumeral joint, depicting abduction/adduction joint motion (G), and the elevation action of the M. deltoideus scapularis (red line) (H). Resting pose of Mussaurus showing the caudoventrally inclined scapular blade, with a caudoventral orientation of the glenohumeral joint, depicting abduction/adduction joint motion (I), and the retraction action of the M. deltoideus scapularis (red line) (J). Note that the movements/actions depicted in both the reference and resting poses are the same (i.e. they are homologous), but differ in the resulting functions performed, because of the reorientation of the glenohumeral joint via scapular reorientation.

Whilst the reference pose was used as a common point of comparison in terms of the DOFs, caution is warranted when one of the studied taxa (in this case Mussaurus) is shifted from the reference pose (with a vertically oriented scapular blade and caudally oriented glenoid) to the resting pose (with a caudodorsally oriented scapular blade and a caudoventral glenoid). Such reorientation of the glenoid (Jenkins, 1993; Gatesy & Baier, 2005) entails drastic modifications of the anatomical and functional implications of the joints’ DOFs (except for pronation/supination). This means that homologous movements in both poses are expressed as different functions in each of them. Consequently, an abduction and adduction movement (i.e. action) is expressed as elevation and depression (i.e. function) in the reference pose but functions as retraction and protraction in the resting pose, respectively (Figs. 4D–4G). This shifting of joint functions must be kept in mind when comparing our results with those of previous work (see Discussion). To minimize confusion and to keep consistency with the Crocodylus model, we conserved the same terms for motions in the reference and the resting poses for Mussaurus, rather than converting the resting pose’s joint motions into different terms. That is, the movement that describes abduction in the reference pose (i.e. the movement parallel to the long axis of the glenoid), was also called abduction in the resting pose, with no reference to the movement relative to the ground (i.e. elevation/depression; protraction/retraction) that the limb would be performing, unless otherwise stated.

Muscle reconstruction

Soft tissue inferences for the myology of Mussaurus patagonicus were established via reference to the literature, by comparisons and homology hypotheses from previous studies of the anatomy of living archosaurs as well as extinct forms (Cooper, 1981; Meers, 2003; Jasinoski, Russell & Currie, 2006; Langer, Franca & Gabriel, 2007; Remes, 2008; Maidment & Barrett, 2011; Burch, 2014; Allen et al., 2014; Klinkhamer et al., 2017) and via additional reference dissections of two specimens of Caiman latirostris (Crocodylia, Alligatoridae) and Gallus gallus (Aves, Galliformes). The extant phylogenetic bracket (EPB; Witmer, 1995) was used to formulate hypotheses about the soft tissue relations in extinct taxa that could be tested by reference to the known osteological correlates of the soft tissues in fossil taxa enclosed by the bracket, constraining speculation to a minimum (Witmer, 1995). We inferred the forelimb muscles’ origin and insertion sites for Mussaurus using this EPB method. Muscle nomenclature used herein is based on Meers (2003) and Burch (2014). A total of 30 muscles were reconstructed in Mussaurus (although some were summed into functional groups for some actions). We avoided reconstructing muscles originating from the body wall (i.e. other than the pectoral girdle or forelimb), including major shoulder muscles such as M. latissimus dorsi, M. pectoralis, M. serratus and others including smaller muscles acting solely on the pectoral girdle (e.g. M. rhomboideus, M. trapezius, M. levator scapulae). Plausible reconstructions of these muscles in the future would require 3D geometry (ideally scanned skeletal material) of the vertebrae, ribs and other elements as well as decisions about the ROMs of any joints therein. M. pronator quadratus, a supinator of the radius-ulna, was not reconstructed in Mussaurus as we did not infer motions within the antebrachium. Similarly, small distal carpus/manus muscles were omitted except where noted below. Muscle abbreviations and EPB levels of inferences are given in Table 1. Our placements of the origin and insertion of each muscle qualitatively approximated the centroids of the estimated areas of attachment inferred from crocodiles and birds (following Hutchinson et al., 2005, 2015). These centroid approximations were used in the next step.

Table 1 Shoulder and forelimb muscles inferred to be present in Mussaurus patagonicus, and their approximate locations.

Muscle	Abbreviation	Origin	Level of inference	Insertion	Level of inference	
Deltoideus scapularis	DS	Lateral surface of the scapular blade	II′	Caudal side of the humerus, close to the humeral head	II	
Deltoideus clavicularis	DC	Acromial region along the craniodorsal surface of the scapula	I′	Caudal surface of the deltopectoral crest	I′	
Teres major	TM	Caudolateral surface of the scapular blade, on the distal half of the blade	II′	Caudal surface of the humerus, medial to the deltopectoral crest	II	
Subscapularis	SBS	Medial surface of the scapular blade, just above the ventromedial ridge	I′	Proximal end of the humerus, medial to the humeral head	I	
Scapulohumeralis posterior	SHP	Caudal margin of the scapular blade, above the scapular glenoid lip	I′	Proximocaudal surface of the humerus, below the humeral head	I′	
Supracoracoideus complex	SC					
 S. longus	SCL	Medial scapula–coracoid boundary	II′	Distal portion of the deltopectoral crest	I	
 S. intermedius	SCI	Lateral scapula–coracoid boundary	I′ or II′	Distal portion of the deltopectoral crest	I or II	
 S. brevis	SCB	Lateral coracoid, above the SCL	I′ or II′	Distal portion of the deltopectoral crest	I or II	
Coracobrachialis brevis dorsalis	CBD	Lateral surface of the scapula, close to the acromion	II′	Proximolateral margin of humerus, above the deltopectoral crest	II′	
Coracobrachialis brevis ventralis	CBV	Lateral coracoid	I′	Internal surface of the deltopectoral crest	I′	
Triceps brachii	
 T. caput scapulare	TBS	Caudolateral surface of the glenoid rim, on scar	I	Ulnar olecranon process	I	
 T. caput coracoideus	TBC	Ramii on the caudal margin of scapula and coracoid	II	Ulnar olecranon process	II	
 T. lateralis	TBL	Caudolateral surface of humeral shaft	I′	Ulnar olecranon process	I	
 T. caput mediale 1	TBM1	Medial and distal portion of the humeral shaft	I′	Ulnar olecranon process	I	
 T. caput mediale 2	TBM2	Caudomedial surface of proximal humerus	I′	Ulnar olecranon process	I	
 T. caput mediale 3	TBM3	Caudal surface of the humeral shaft	II′	Ulnar olecranon process	II	
 T. caput mediale 4	TBM4	Lateral and distal portion of the humeral shaft	II′	Ulnar olecranon process	II	
Biceps brachii	BB	Craniodorsal surface of the coracoid	I′	Proximomedial surface of the radius	I	
Humeroradialis	HR	Craniodorsal surface of humerus, caudal to the deltopectoral crest	II′	Humeroradialis tubercle of the proximal radius, on craniolateral side	II	
Brachialis	BR	Craniomedial surface of the humerus, distal to the deltopectoral crest, or on the cuboid fossa	I or I′	Proximomedial surface of the radius	I′	
Supinator	SU	Ectepicondyle of the humerus	I	Cranial radial shaft	I′	
Extensor carpi radialis	ECR	Ectepicondyle of the humerus	I	Dorsal surface of distal carpal I	II′	
Extensor carpi ulnaris	ECU	Ectepicondyle of the humerus	II	Dorsolateral surface of metacarpal II	II′	
Flexor ulnaris	FU	Ectepicondyle of the humerus	I	Craniolateral surface of ulna	I′	
Abductor radialis	AR	Ectepicondyle of the humerus	II	Cranial surface of the radius	II′	
Pronator teres	PT	Entepicondyle of the humerus	I′	Proximomedial surface of radius	I′	
Abductor pollicis longus	APL	Lateral shaft of the radius and ulna	I′	Proximomedial margin of metacarpal I	II′	
Extensor digitorum longus	EDL	Ectepicondyle of the humerus	I	Proximodorsal margin of metacarpal II	I′	
Extensor digiti I superficialis	EDS	Distal and anterior surface of radius and ulna and probably distal carpal I	I′	Extensor process of ungual phalanx	I′	
Extensor digiti I profundus	EDP	Dorsolateral and dorsodistal surface of metacarpal I	I′	Extensor process of ungual phalanx	I′	
Flexor digitorum brevis superficialis digiti I	FDSI	Distal carpals	II	Flexor processes of phalanx I	II	
Flexor digitorum profundus digiti I	FDPI	Distal carpals	I	Flexor process of phalanx I	I	
Flexor digitorum longus	FDL	Entepicondyle of the humerus, caudal surface of the ulna, and ulnar surface of distal carpals	I′	Flexor surface of ungual phalanges	I	
Note:

Levels of inference correspond to those that are conservative in extant archosaurs (I) or varied and thus ambiguous for Archosauria (II); level III inferences (parsimoniously absent in ancestral Archosauria) were not used. Prime (I′, II′) annotations indicate attachments lacking clear osteological correlates, which can still be reconstructed but only have approximate, relative rather than more specific, direct locations (I, II).

Muscle path specification

Once muscles were positioned at their respective origins and insertions, the next step was to model plausible paths over which each muscle would move during motion of the joints. Otherwise, a uniformly straight line of action of muscles would create unnatural paths, crossing over (or through) the bones or other muscles in implausible ways, resulting in dubious moment arms as outputs. We used ‘via points’ and ‘wrapping surfaces’ to create anatomically realistic paths. Via points are fixed points attached to a body segment that can be used to implement simple constraints on a muscle’s path relative to a bone or other structure (Fig. S1). For example, the triceps muscle group, originating on the scapula/coracoid and the humeral shaft, needed via points to avoid the assumed shape that the more internally located muscles might have had, as exemplified by M. triceps brachii caput mediale 1 (TBM1) (internally placed) and M. triceps brachii caput scapulare (TBS) (externally placed).

A wrapping object (or surface) is a geometric form (Fig. S2), also associated with a body segment, which is assigned to one or more muscle(s) and creates a deflection of their path when crossed, preventing any associated muscle from penetrating it (Delp et al., 1990; see also Hutchinson et al., 2005, 2015). Wrapping objects’ attributes are listed in Table S1.

Most wrapping objects were represented as cylinders, used to represent physical bone surfaces, to constrain muscle paths, and to imitate unpreserved attributes (e.g. cartilage). This latter point is very important because a large amount of articular cartilage is missing in extinct reptiles, affecting the paths of muscles involved (Hutchinson et al., 2005). The elbow joint is critical because our inferences of its morphology would be affected by missing articular cartilage (Fujiwara, Taru & Suzuki, 2010; Holliday et al., 2010) and the main elbow (and other distal) extensor and flexor musculature would pass closely around this joint, with their paths influenced by this cartilage. A set of cylinders, serving as wrapping surfaces for one or more muscles, was placed parallel to the humeral condylar axis at varied distances from the condyles (see sensitivity analysis of moment arms below and in ‘Discussion’), taking the role of the articular cartilage on constraining muscle paths around the elbow.

Considering that our Mussaurus model exhibited differences in joint orientations between the reference and the resting pose, and the former was actually an implausible pose for a basal sauropodomorph, some muscle paths required additional constraints to fit the reference pose (Fig. S3; Table S1). Our complete models for the forelimbs of Crocodylus and Mussaurus are available online (Otero et al., 2017a, 2017b) at https://figshare.com/articles/Crocodylus_musculoskeletal_models/4928696 and https://figshare.com/articles/Mussaurus_musculoskeletal_models/4928684.

Joint ROM analysis

Analyses of the forelimb joints’ ROM were conducted for both the reference and resting poses, for each DOF allowed for each joint in the model. Estimation of ROM was done with the musculoskeletal software, by manipulating each DOF manually and visualizing in 3D at which joint angles the bones came into close proximity and thus would pass through each other (or their presumed cartilage) if moved further (Pierce, Clack & Hutchinson, 2012; Reiss & Mallison, 2014; Arnold, Fischer & Nyakatura, 2014). ROM estimation was not performed directly from bone to bone surfaces, but rather left 10% space of total bone length distributed among the glenohumeral, elbow and wrist joints for emulation of the cartilage volume that might have existed in life (following Holliday et al., 2010). Thus, our ROM analysis was roughly equivalent to ‘ROM4’ of Hutson & Hutson (2012) in which all soft tissues but cartilage was removed. Considering that the reference pose was not a realistic posture, we expected that ROM values estimated from a resting pose would be smaller because of the restrictions imposed by more realistic movements of the joints.

The reference pose in this study represented how the joint was positioned at zero degrees; thus, any angular rotation away from that pose would be either positive or negative. To clarify further, when the models were positioned in the resting pose (or any other pose), the 0° position was not altered, and any other positions of the limb segments were quantified relative to that 0° reference pose.

Analysis of muscle moment arms

We calculated muscle moment arms about the glenohumeral, elbow, and wrist joints for the Mussaurus and Crocodylus models (in SIMM software, Delp et al., 1990; Delp & Loan, 1995). Additionally, we explored muscle moment arms in extension/flexion for manual digit I in Mussaurus because this digit in early sauropodomorphs has a medially deflected claw that has been hypothesized as having played a key role in manus functions other than locomotion, such as grasping and browsing (Galton & Upchurch, 2004; Yates et al., 2010).

For the glenohumeral joint, moment arms were calculated for all three rotational DOFs, considering that movements of the humerus allowed for appreciable amounts of extension/flexion, abduction/adduction, and pronation/supination in both Mussaurus and Crocodylus (see ROM below; additional plots are in the Online Supplementary Text as per the Results). For the remaining joints (i.e. elbow, wrist, MCP and INP), only flexion and extension moment arms were calculated because this DOF corresponded to the main axis around which those joints predominantly would act, and this simplified our analysis (but see Discussion for the elbow joint).

Moment arms were first calculated in the Mussaurus and Crocodylus models for the reference pose. If a muscle had a certain action for more than 75% of a given DOF’s ROM, it was plotted as a ‘pure’ action muscle (e.g. ‘pure’ flexor), otherwise it was plotted as ‘mixed.’ If there was a mismatch between the taxa (e.g. a muscle being a flexor in Crocodylus but an extensor in Mussaurus), then we also plotted that muscle in the ‘mixed’ action category. We also calculated the moment arms for the Mussaurus model in the resting pose for comparison with the reference pose.

Moment arm values vary depending on the paths of muscles (Delp et al., 1990; Hutchinson et al., 2005), so alteration of either origin or insertion sites (as well as paths influenced by wrapping or via points) may affect moment arm estimations, and possible muscle actions. Here we focused our sensitivity analysis on the elbow of Mussaurus, for which the articular cartilage volume is unknown and likely was considerable (Schwarz, Wings & Meyer, 2007). We started with a minimum amount of cartilage, increased the elbow cap to a maximum, and then evaluated how extensor muscle moment arms were affected by these assumptions (see Discussion). Here, the minimum and maximum amounts of cartilage were defined taking as a reference the standardized 10% of the total limb length previously proposed in the literature for archosaurs (Holliday et al., 2010) and then we added and removed cartilage accordingly.

Our presentation of moment arm values required some normalization to facilitate comparisons between Crocodylus and Mussaurus, because these taxa differ so greatly in body size and forelimb morphology. Following typical practice (Hutchinson et al., 2008; Bates & Schachner, 2012), we normalized the moment arms by corresponding segment lengths (humerus, antebrachium (radius-ulna) and metacarpal II for shoulder, elbow and distal joints; data in Table S2), using the distances between joint centres in the model as the lengths. However, as forelimb proportions clearly changed between Archosauria and these two taxa, segment lengths are not the ideal metrics for normalization. We consider this problem in the ‘Discussion.’

Results

Muscle reconstruction

Non-avian archosaurs represent a particular challenge when reconstructing forelimb musculature based on an extant phylogenetic bracketing framework because of deep functional disparities, related to the different modes of locomotion existing between extinct and the living forms (e.g. sprawling vs. erect; biped vs. quadruped; non-flying vs. flying). Although the inferences of presence/absence of the forelimb musculature reconstructed herein for Mussaurus (Fig. 3) were based on the EPB approach (Witmer, 1995), our final decisions of muscle position and extent (e.g. in equivocal cases; Level II′ inferences) were based mainly on extant Crocodylia because of the greater morphological similarities that this group shares with non-avian dinosaurs than with birds (Jasinoski, Russell & Currie, 2006; Remes, 2008; Burch, 2014; Maidment et al., 2014).

Within the shoulder musculature, one important difference from previous contributions is the reconstruction of the M. teres major (TM) in Mussaurus. This muscle is absent in most sauropsids (Remes, 2008) and was reconstructed neither in theropods (Nicholls & Russell, 1985; Jasinoski, Russell & Currie, 2006; Burch, 2014) nor in basal ornithischians (Maidment & Barrett, 2011). However, the TM is present in extant Crocodylia (Meers, 2003; Allen et al., 2014; Klinkhamer et al., 2017), thus representing a Level II inference for an insertion on the humerus, just medial to the deltopectoral crest, on a proximodistally elongated crest. The TM muscle was also inferred to have been present in the basal sauropodomorphs Saturnalia and Efraasia (Remes, 2008).

The origin and insertion of M. deltoides clavicularis (DC) are rather congruent among different studies, taking origin from the acromial area of the scapula and inserting on the lateral aspect of the deltopectoral crest of the humerus in both Crocodylia and non-avian Dinosauria (Meers, 2003; Jasinoski, Russell & Currie, 2006; Suzuki & Hayashi, 2010; Maidment & Barrett, 2011; Burch, 2014; Klinkhamer et al., 2017). However, an origin from the clavicle was reported in lepidosaurs (Russell & Bauer, 2008) and birds (Burch, 2014), but clavicles are only known for the basal sauropodomorphs Massospondylus, Plateosaurus (Yates & Vasconcelos, 2005) and Adeopapposaurus (Martínez, 2009). Considering that there is no evidence of clavicles in Mussaurus, such an origin site was not reconstructed here. Remes (2008), on the other hand, proposed the presence of clavicles throughout sauropodomorph evolution, and that these were the osteological correlate for the DC. Regardless, the origin site and the line of action of this muscle would not be drastically affected by the presence of clavicles as reconstructed by Yates & Vasconcelos (2005) or Remes (2008).

The coracobrachialis (CB) muscle was reconstructed in Mussaurus with a single head (M. coracobrachialis brevis), and two divisions of that (pars ventralis and dorsalis; M. coracobrachialis brevis ventralis (CBV) and M. coracobrachialis brevis dorsalis (CBD)) as in living crocodiles (Meers, 2003; Suzuki & Hayashi, 2010; Allen et al., 2014; Klinkhamer et al., 2017). There are two heads for this muscle (pars cranialis and caudalis) in extant birds, both originating from the craniolateral aspect of the coracoid (Vanden Berge & Zweers, 1993). Based on its anatomical position, the M. coracobrachialis cranialis of birds should be equivalent to the CBV of Crocodylia, and it would insert on the base of the deltopectoral crest of the humerus (Vanden Berge, 1982). An additional head, M. coracobrachialis longus, was reported as absent in Crocodylia (Jasinoski, Russell & Currie, 2006; Remes, 2008; but see Nicholls & Russell, 1985) and present in some birds (Jasinoski, Russell & Currie, 2006). Langer, Franca & Gabriel (2007), however, inferred the presence of this muscle in Saturnalia, taking into account that most neognaths have it, although we do not agree with such an inference considering the drastic modifications of avian forelimbs (i.e. a level II inference for Mussaurus) so we did not reconstruct the M. coracobrachialis longus in Mussaurus.

The supracoracoideus (SC) muscle has two heads in Alligator mississippiensis (Meers, 2003). In extant birds there is a single head, but with multiple origins (e.g. keel, mesosternum, manubrium, Vanden Berge & Zweers, 1993; Jasinoski, Russell & Currie, 2006). Homologies with Crocodylia are controversial, with no consensus on whether the scapular (Remes, 2008) or coracoid (Maidment & Barrett, 2011) head was lost in birds. The origin site (either single or multiple) of the SC complex is consistently located around the scapula–coracoid boundary in Crocodylia, and always inserts on the deltopectoral crest. Thus for the biomechanical purposes of this study, we reconstructed the SC in Mussaurus as a single head originating from the centroid of the area where any head(s) should have originated. In addition, that area is not preserved in any of the Patagonian specimens, precluding the identification of osteological correlates for the origin of this muscle.

The scapulohumeralis was reconstructed in Mussaurus as a single head (M. scapulohumeralis posterior, SHP), corresponding to the M. scapulohumeralis caudalis of Crocodylia (Meers, 2003; Remes, 2008; Suzuki & Hayashi, 2010; Allen et al., 2014; Klinkhamer et al., 2017). Scapulohumeralis anterior was not reconstructed in Mussaurus because it is absent in Crocodylia (Meers, 2003; Jasinoski, Russell & Currie, 2006; Suzuki & Hayashi, 2010; Burch, 2014), although it is reported in birds. The medial side of the scapula of Mussaurus has a long ridge extending parallel to both margins (ventromedial ridge, Otero & Pol, 2013), which has been hypothesized as the boundary of SHP (ventrally) and SBS (dorsally) (Langer, Franca & Gabriel, 2007; Burch, 2014).

The inferred number of heads of the Mm. triceps brachii (TB) for archosaurs are four (Jasinoski, Russell & Currie, 2006; Burch, 2014) or five (Meers, 2003; Remes, 2008). In Mussaurus, as in extant Crocodylia we infer that there were two origin sites from the scapulocoracoid (i.e. TBS, triceps caput coracoideus (TBC)) and two from the humeral shaft (i.e. triceps brachii lateralis (TBL), triceps brachii caput mediale (TBM)). Regardless of all controversies surrounding the precise number of humeral heads in living archosaurs, for our purposes of muscle moment arm analysis and considering the lines of action of this large muscle group, the reconstruction of the humeral head in Mussaurus (TBM) was split into four portions, which corresponded to the different areas on the humeral shaft from which the TBM probably originated. Previous reconstructions of M. triceps in dinosaurs vary. The triceps group was reconstructed with two scapulocoracoid heads and one humeral head in Saturnalia (Langer, Franca & Gabriel, 2007) and only two heads in early ornithischian dinosaurs (one from the scapulocoracoid and one from the humerus; Maidment & Barrett, 2011). Within later ornithischians, five heads were inferred in Euoplocephalus (Coombs, 1978), corresponding to those described for extant crocodiles.

The origin of the biceps brachii (BB) mucle in extinct forms is equivocal: some studies place it just cranially to the glenoid lip of the coracoid (Langer, Franca & Gabriel, 2007; Remes, 2008; Burch, 2014), whereas the origin in Crocodylia is even more cranially placed, close to the scapulocoracoid boundary (Meers, 2003; Suzuki & Hayashi, 2010), a hypothesis followed here (see also Maidment & Barrett, 2011). A second head of this muscle on the humerus (as present in some birds, Remes, 2008) is too speculative (Level II′) because it is absent in Crocodylia (Meers, 2003) and no corresponding scars are evident in Mussaurus.

The brachialis (BR) and humeroradialis (HR) muscle attachments seem to retain their ancestral origins and insertions in most sauropsids, with some secondary changes in birds. In Mussaurus we infer that they originated from the humeral shaft, close to the deltopectoral crest, and inserted on the proximomedial surface of the radius (Jasinoski, Russell & Currie, 2006; Remes, 2008; Burch, 2014). However, Cooper (1981), Langer, Franca & Gabriel (2007) and Maidment & Barrett (2011) placed the BR origin more distally, as in birds; a conclusion that we deem to be less convincing (Level II′ inference).

Most of the muscles originating from the humeral condyles and inserting either on the radius or ulna, such as Mm. supinator (SU), flexor ulnaris (FU) (anconeus sensu Burch, 2014), and pronator teres (PT) do not exhibit major differences between extant Archosauria. Therefore, qualitative reconstructions in extinct forms remain unequivocal (Remes, 2008; Burch, 2014; but see Langer, Franca & Gabriel, 2007), with the exception of the abductor radialis (AR), which is not present in birds.

The M. extensor digitorum longus (EDL; extensor carpi ulnaris longus sensu Meers, 2003) of sauropsids has an insertion that varies between the dorsal proximal portions of the metacarpals, depending on the group (Remes, 2008; Burch, 2014). Insertions onto the bases of metacarpals I and II are phylogenetically unequivocal for Mussaurus. Considering that reconstructions of the insertion onto MCI, MCII or both would not appreciably affect the EDL’s line of action as it crosses the wrist joint, we reconstructed this muscle as inserting only onto MCII.

Remes (2008) inferred that sauropodomorphs lacked M. extensor carpi ulnaris (ECU), but this is contradicted by evidence from crocodylian myology (dozens of specimens studied by Meers (2003) and Allen et al. (2014) so our models incorporated the ECU for Crocodylus and Mussaurus (also see discussion in Burch (2014, 2017) and Klinkhamer et al. (2017). The archosaurian ECR (M. extensor carpi radialis longus sensu Meers, 2003) muscle inserts onto the radiale. Considering that basal sauropodomorphs lack a preserved radiale, we reconstructed the ECR as inserting onto the distal carpus in Mussaurus (as in Aves). Similarly, the origin site of the abductor pollucis longus (APL; M. extensor carpi radialis brevis sensu Meers, 2003) was placed on the lateral side of the radius in Mussaurus. The osteological correlate of this origin was assessed to be a small tubercle on the lateral and distal area of the radius, also reported in Saturnalia (Langer, Franca & Gabriel, 2007). An additional origin from the lateral ulna (Remes, 2008; Burch, 2014) was omitted because it would have had the same general line of action and hence would not affect the action of this muscle in Mussaurus.

The flexor digitorum longus (FDL) muscle ancestrally had humeral, ulnar, and carpal heads, all of them joining into a single tendon that then diverged to insert on the manual digits (Meers, 2003). As one single common tendon passes across the wrist, we reconstructed only the humeral head for Mussaurus, inserting on the flexor surfaces of the manual phalanges (i.e. proximoventral aspect).

Nomenclature for the extensor and flexor musculature of the digits remains controversial among living sauropsids, especially considering the extensive modifications of the avian forelimb. Thus we withheld from reconstructing these muscles in detail for Mussaurus, simply following the scheme from Meers (2003). Muscles extensores digitorum superficiales (EDS) and profundus (EDP) in Mussaurus both originated from the distal aspect of metacarpal I (for a similar myology, see Burch (2014): extensores digitores breves, EDB), although leaving no muscle scars. In extant birds, the EDP’s putative equivalent by position would be the M. extensor brevis alulae, but originating from the extensor apophysis of the metacarpus and the alula (Vanden Berge, 1982). Hence a Level II inference resulted for this muscle in Mussaurus, and we applied the character state observed in Crocodylia rather than in Aves.

The flexores digitorum superficialis (FDS) and profundus (FDP) muscles originate proximally from the distal carpals and distally insert onto the flexor process of the first phalanx of the digits in Crocodylia (Meers, 2003; Burch, 2014). As with the extensor musculature of the digits, avian homologues are difficult to establish, but judging from its position the M. flexor alulae is a plausible candidate (Vanden Berge, 1982; Burch, 2014). We applied crocodylian myology to Mussaurus.

Joint ROM analysis

Here we consider the results of the ROM analysis for Mussaurus patagonicus in the resting pose, whereas for Crocodylus johnstoni, ROMs for the reference pose were the same as those for the resting pose (Table 2; Table S3).

Table 2 Ranges of motion (ROMs) of each degree of freedom for Mussaurus and Crocodylus in the resting pose.

	Joint	Pronation (°)	Supination (°)	Total long-axis rotation (°)	Abduction (°)	Adduction (°)	Total ab/adduction (°)	Flexion (°)	Extension (°)	Total flexion/extension (°)	
Mussaurus patagonicus	Glenohumeral	−25	25	50	−25	40	65	−35	−70	35	
Elbow	−30	0	30	−5	5	10	130	20	110	
Wrist	–	–	–	−10	10	20	70	−30	100	
Metacarpo-phalangeal	–	–	–	–	–	–	50	−40	90	
Interphalangeal	–	–	–	–	–	–	70	−25	95	
Crocodylus johnstoni	Glenohumeral	−20	20	40	−5	45	50	5	−60	65	
Elbow	−20	8	28	−5	5	10	110	15	95	
Wrist	−10	30	40	−30	5	35	40	−60	100	

Pronation and supination values around the glenohumeral joint had similar values for Mussaurus (−25°/25°) and Crocodylus (−20°/20°), for a total maximal ROM of 50° and 40°, respectively. The glenohumeral joint of Mussaurus allowed −25° of abduction and 40° of adduction for a total ROM of 65°. On the contrary, the Crocodylus model showed a reduced capacity for abduction from the reference pose (−5°) and allowed 45° of adduction, for a total glenohumeral ROM of 50°. The flexion and extension axis of the glenohumeral joint in Mussaurus allowed −35° of flexion and −70° of extension from the 0° pose, for a total ROM of 35°. Crocodylus showed a greater glenohumeral ROM of 65° (5° flexion to −60° extension).

Long axis rotation at the elbow showed interesting values in the Mussaurus model, allowing −30° of pronation and 0° of supination, for a total ROM of 30°. In Crocodylus, less pronation than in Mussaurus was allowed (−20°) starting from the reference pose, but more supination as well (8°), for a total of 28° of long axis rotation. Abduction and adduction, on the other hand, showed no differences between both Mussaurus and Crocodylus, allowing a total ROM of 10° (5° in each direction). Finally, flexion and extension at the elbow showed similar ROM values in Mussaurus and Crocodylus, allowing flexion to 130° in Mussaurus and 110° in Crocodylus. Extension values could not surpass 20° in Mussaurus and 15° in Crocodylus.

Pronation and supination at the wrist were precluded in Mussaurus. Abduction and adduction showed the same values for Mussaurus (−10°/10°), whereas Crocodylus had more abduction capacity (−30°) in contrast to adduction (5°). Flexion and extension of the wrist joint was 70° and −30°, respectively for Mussaurus, whereas in Crocodylus flexion showed smaller ROM values (40°) and extension ROM was greater (−60°) than in Mussaurus. The MCP joint of digit I in Mussaurus had 50° of flexion and −40° of extension ROM, whereas the INP joint allowed the angle of flexion to increase to 70° and extension could reach −25°.

Muscle moment arm analysis

Here we compare the moment arm values obtained for Crocodylus and Mussaurus in the resting pose. For the glenohumeral joint, moment arms were measured while varying the degree of freedom they relate to (pronation/supination, extension/flexion, and abduction/adduction) to quantify how joint orientation affected muscle action. We also plotted glenohumeral pronation/supination and abduction/adduction moment arms against glenohumeral extension/flexion angle (see Online Supplementary Text). For the elbow and wrist, we focused on extension/flexion moment arms and joint angles because this is the main motion performed by those joints. Then we focus on inferences about broad trends in muscle actions (and, where feasible, general functions) inferred from the resting pose.

Glenohumeral joint (Figs. 5–7; Figs. S8 and S9; Table 3; Table S4)

Some muscles showed similar actions for pronation and supination around the glenohumeral joint in Crocodylus and Mussaurus, whereas others displayed differences in muscle action between taxa (Fig. 5). Muscles originating from the scapular blade and inserting well lateral or well medial on the proximal humerus had the same action for both taxa; e.g. some humeral supinators such as DC, DS and TM. TBC and SC consistently were supinators, although the TBC inserted on the olecranon process of the ulna rather than on the humerus in both taxa, and the SC originated on the proximal scapula and coracoid and not on the scapular blade. The SBS and CBV remained as humeral pronators for both taxa (Fig. 5A).

Figure 5 Pronation/supination moment arms around the glenohumeral joint, normalized to humerus segment length, plotted against pronation/supination joint angles for Crocodylus and Mussaurus in the resting pose.

(A) Mostly pronators; (B) mostly supinators; (C) mixed pronators/supinators. Negative moment arms and glenohumeral angles correspond to pronation, while positive values correspond to supination. For muscle abbreviations, see Table 1. For a similar plot using glenohumeral extension/flexion joint angle on the x-axis, see Fig. S8.

Figure 6 Extension/flexion moment arms around the glenohumeral joint, normalized to humerus segment length, plotted against extension/flexion joint angles for Crocodylus and Mussaurus in the resting pose.

(A) Mostly extensors; (B) mostly flexors; (C) mixed extensors/flexors. Negative moment arms and glenohumeral angles correspond to extension, while positive values correspond to flexion. For muscle abbreviations see Table 1.

Figure 7 Abduction/adduction moment arms around the glenohumeral joint, normalized to humerus segment length, plotted against abduction/adduction joint angles for Crocodylus and Mussaurus in the resting pose.

(A) Mostly abductors; (B) mostly adductors; (C) mixed abductors/adductors. Negative moment arms and glenohumeral angles correspond to abduction, while positive values correspond to adduction. For muscle abbreviations see Table 1. For a similar plot using glenohumeral extension/flexion joint angle on the x-axis, see Fig. S9.

Table 3 Muscle actions for the glenoid and elbow joints for Crocodylus and Mussaurus in the resting pose.

Muscle	Glenoid	Elbow	
Pronation/supination	Extension/flexion	Abduction/adduction	Extension/flexion	
Crocodylus	Mussaurus	Crocodylus	Mussaurus	Crocodylus	Mussaurus	Crocodylus	Mussaurus	
DS	Supination	Supination	Flexion	Flexion	Abduction	Abduction	–	–	
DC	Supination	Supination	Flexion	Flexion	Abduction	Abduction	–	–	
TM	Supination	Supination	Extension	Extension	Abduction	Abduction	–	–	
SBS	Pronation	Pronation	Extension	Extension	Adduction	Adduction	–	–	
SHP	Supination	Pronation	Extension	Extension	Abduction	Abduction	–	–	
SCI	Supination	Supination	Flexion	Flexion	Adduction	Adduction	–	–	
SCB	Supination	Supination	Flexion	Flexion	Adduction	Adduction	–	–	
SCL	Supination	Supination	Flexion	Flexion	Adduction	Adduction	–	–	
CBV	Pronation	Pronation	Extension	Flexion	Adduction	Adduction	–	–	
CBD	Mixed	Supination	Flexion	Flexion	Adduction	Mixed	–	–	
TBS	Supination	Mixed	Extension	Mixed	Abduction	Abduction	Extension	Extension	
TBC	Supination	Supination	Extension	Extension	Abduction	Abduction	Extension	Extension	
TBM4	–	–	–	–	–	–	Extension	Extension	
TBM1	–	–	–	–	–	–	Extension	Extension	
TBM3	–	–	–	–	–	–	Extension	Extension	
TBL	–	–	–	–	–	–	Extension	Extension	
TBM2	–	–	–	–	–	–	Extension	Extension	
BB	Supination	Mixed	Flexion	Flexion	Adduction	Adduction	Flexion	Flexion	
HR	–	–	–	–	–	–	Flexion	Flexion	
BR	–	–	–	–	–	–	Flexion	Flexion	
SU	–	–	–	–	–	–	Extension	Mixed	
FU	–	–	–	–	–	–	Extension	Mixed	
AR	–	–	–	–	–	–	Extension	Mixed	
PT	–	–	–	–	–	–	Extension	Flexion	
FDL	–	–	–	–	–	–	Mixed	Flexion	
EDL	–	–	–	–	–	–	Extension	Mixed	
ECR	–	–	–	–	–	–	Extension	Mixed	
ECU	–	–	–	–	–	–	Extension	Mixed	
Note:

Bold font highlights a difference between the two taxa. ‘–’ indicates that the muscle was inferred not to act around that axis in the model.

In contrast, BB, SHP, TBS and CBD had variable actions that differed between Mussaurus and Crocodylus (Fig. 5C). SHP acted as a supinator in Crocodylus, but was a pronator in Mussaurus; BB and TBS were fully supinators in Crocodylus, but had a mixed action in Mussaurus. Finally, CBD had very different actions in the two taxa: in Crocodylus it switched between weak pronation/supination whereas in Mussaurus it was consistently a strong supinator.

Within the category of pronator/supinator actions, muscles showed contrasting patterns in Crocodylus and Mussaurus for moment arm magnitudes around the glenohumeral joint, especially as joint orientation was varied between supination and pronation (Figs. 5B and 5C). For example, most supinator muscles in Crocodylus (DC, SHP, TM, SC, TBS, TBC and BB) experienced an increase of their moment arms with pronation. In contrast, in Mussaurus, only two (SC and TBC) of the six supinator muscles increased their supinator moment arms with pronation. The remaining glenohumeral supinators in Mussaurus (DS, DC, TM and CBD) displayed patterns of increasing supinator moment arms with glenohumeral supination, not pronation. Pronator muscles (e.g. CBV), however, displayed similar patterns in both taxa (Fig. 5A), showing a general increase of pronator moment arms with supination; with the exception of the SBS muscle in Mussaurus, which exhibited almost constant large moment arms.

SBS, SHP, TBC and TM were extensors in both taxa (Fig. 6A), as well as TBS in Crocodylus. On the other hand, the BB, CBD, SC, DS and DC muscles, which originated from the cranial surface of the scapulocoracoid, were shoulder flexors in both taxa (Fig. 6B).

The CBV and TBS muscles had strikingly different actions in the two taxa. CBV was mainly a weak shoulder extensor in Crocodylus (shifting from extension to flexion at about −15°) but it had a strong, consistent flexor action action in Mussaurus (Fig. 6C). TBS, in contrast, consistently had a strong shoulder extensor action in Crocodylus but was a mainly a weak flexor in Mussaurus (shifting to extensor action at about −65°).

The only muscle that increased its extensor moment arm with glenohumeral joint flexion in both taxa was SBS. The remaining extensor muscles exhibited different patterns (Fig. 6A). In Crocodylus, all flexor muscles increased their moment arm about the glenohumeral joint with flexion, except for the CBD, which displayed less change (Fig. 6B). In Mussaurus, DS, DC and CBV also increased their flexor moment arms with joint flexion, whereas SC and BB showed an increase of flexor moment arms with joint extension.

We found that muscles acting about the glenohumeral abductor/adductor axis displayed similar patterns in Crocodylus and Mussaurus. Many muscles (DC, DS, SHP, TBC, TBS and TM) were glenohumeral abductors in both Crocodylus and Mussaurus (Fig. 7A). The BB, CBV, SBS and SC muscles, however, had adduction actions for both taxa (Fig. 7B), whereas CBD acted as an adductor in Crocodylus and as a mixed abductor/adductor in Mussaurus (Fig. 7C).

The glenohumeral abductor muscles DC, DS and TBS increased their moment arm with joint abduction in Crocodylus, whereas all abductor muscles (DC, DS, SHP, TBC, TBS and TM) in Mussaurus showed the same pattern (Fig. 7A). In Crocodylus, most adductor muscles (CBD, CBV, BB and SC) increased their moment arms with glenohumeral joint adduction, except for SBS and CBD, which had the opposite pattern (Figs. 7B and 7C). In Mussaurus, BB and SC increased their adduction moment arms with humeral abduction, whereas SBS and CBV remained rather constant and only CBD increased its adduction moment arm with joint adduction (becoming an adductor as the shoulder was adducted past −10°; Fig. 7C).

While the above actions generally held across their same DOFs (e.g. pronation/supination moment arms vs. joint angles), there were some interesting variations when other DOFs were plotted. Pronator/supinator actions varied with glenohumeral extension/flexion angle (Fig. S8), showing that while the SBS and CBV retained the same actions across this DOF, other muscles such as the SC group became more mixed in their actions especially in Mussaurus—and despite its limited glenohumeral ROM around that axis. TBC and BB even switched entirely to pronators in Mussaurus, and TBS became more mixed action than just a supinator in Crocodylus. Moreover, abductor/adductor actions shifted slightly when plotted the same way—most noticeably, the CBD and SBS muscles switched positions (cf. Figs. 7B and 7C vs. Figs. S9B and S9C), with Mussaurus and Crocodylus both having consistent adductor actions for the CBD whereras the SBS stayed solely an adductor in Mussaurus but took on a more mixed action in Crocodylus.

Elbow joint (Fig. 8; Table 3; Table S5)

Although elbow adduction and abduction occur during ‘sprawling’ locomotion in Alligator (Fujiwara & Hutchinson, 2012; Baier & Gatesy, 2013) and we allowed three DOFs at this joint in both models, here we only consider flexion and extension for the purpose of our moment arm analyses (but see below), because we expect elbow extension/flexion to have predominated in Mussaurus. Generally, muscle actions around the elbow showed fewer differences between Crocodylus and Mussaurus compared to the glenohumeral joint. In addition, the major elbow extensors and flexors had similar patterns in both taxa, although varying in their relative moment arm magnitudes.

Figure 8 Extension/flexion moment arms around the elbow joint, normalized to radius-ulna segment length, plotted against extension/flexion joint angles for Crocodylus and Mussaurus in the resting pose.

(A) Mostly extensors; (B) mostly flexors; (C) mixed extensors/flexors. Negative moment arms correspond to extension, while positive values correspond to flexion. Zero elbow angle corresponds to full extension, while larger angles correspond to flexion. For muscle abbreviations see Table 1.

The triceps group includes the main elbow extensor muscles. In Crocodylus and Mussaurus, these muscles maintained a similar pattern of action (Fig. 8A), with smaller values at full extension, mostly increasing their extensor moment arms as the elbow was flexed. The most noteworthy difference between the action of the triceps group in both taxa was that the moment arm value in Mussaurus substantially increased (almost twice as large) between full extension to full flexion, whereas in Crocodylus the values at maximal flexion and extension did not differ greatly. In both taxa, the peak values of moment arms occurred neither at full extension nor full flexion, but at moderate elbow joint angles (∼60–100°).

Elbow flexor muscles revealed similar patterns in Crocodylus and Mussaurus, generally increasing their flexor moment arm with increasing joint flexion (Fig. 8B). The BB, BR and HR all reached peak flexor moment arms (about two times the minimal values) at moderate elbow flexion angles (∼90°). In contrast, the PT and FDL only acted as weak elbow flexors in Mussaurus, with minimal changes of their moment arms.

The remaining muscles acting around the elbow joint corresponded to those originating on the distal humeral condyles. We found that, in Mussaurus, most of these antebrachial muscles (AR, SU, ECR, ECU, EDL and FU) shifted from flexor to extensor moment arms as the elbow became more flexed (between 55° and 65°; Fig. 8C). One interesting difference observed between Crocodylus and Mussaurus was that, apart from the triceps group, different muscles acted as elbow extensors in the crocodile (AR, SU, ECR, EDL, PT and FU); FDL being the only mixed-action elbow muscle in Crocodylus (but a consistent flexor in Mussaurus). The FU muscle in Crocodylus had the largest moment arm and remained an extensor throughout its ROM; unlike in Mussaurus. Likewise, the PT muscle was always an elbow extensor in Crocodylus but a flexor in Mussaurus. Overall, more muscles acting around the elbow of Mussaurus tended to act as flexors, compared to the pattern in Crocodylus.

Wrist and manus joints (Figs. 9–11; Table 4; Tables S6 and S7)

Muscles acting around the distal forelimb joints in Crocodylus and Mussaurus showed minimal switches of action; only in some cases switching at extremes of extension/flexion ROM (Figs. 9–11). General patterns of moment arm changes with joint angles did not differ remarkably, either. In both taxa, there were multiple carpal extensors including the ECR, ECU, APL and EDL (Fig. 9A). Most of these muscles increase their moment arm with joint flexion (except for the EDL in Crocodylus, which maintained almost constant extreme values). Similarly, the FDL was a carpal flexor in both taxa, increasing flexor moment arm with joint extension (Fig. 9B).

Figure 9 Extension/flexion moment arms around the wrist, normalized to metacarpal II length, plotted against extension/flexion joint angles for Crocodylus and Mussaurus.

(A) Mostly extensors; (B) mostly flexors. Negative moment arms and wrist joint angles correspond to extension, while positive values correspond to flexion. For muscle abbreviations see Table 1.

Figure 10 Extension/flexion moment arms (not normalized) around the metacarpo-phalangeal joints (MCP), plotted against extension/flexion joint angles for Mussaurus.

(A) Mostly extensors; (B) mostly flexors. Negative moment arms and joint angles correspond to extension, while positive values correspond to flexion. For muscle abbreviations see Table 1.

Figure 11 Extension/flexion moment arms (not normalized) around the interphalangeal (INP) joints, plotted against flexion/extension joint angles for Mussaurus.

(A) Extensors; (B) flexors. Negative moment arms and joint angles correspond to extension, while positive values correspond to flexion. For muscle abbreviations see Table 1.

Table 4 Muscle actions for the wrist and manus of Crocodylus and Mussaurus.

Muscle	Wrist	Metacarpo-phalangeal	Interphalangeal	
Extension/flexion	Extension/flexion	Extension/flexion	
Crocodylus	Mussaurus	Mussaurus	Mussaurus	
ECR	Extension	Extension	–	–	
APL	Extension	Extension	–	–	
EDL	Extension	Extension	–	–	
ECU	Extension	Extension	–	–	
EDS	–	–	Extension	Extension	
EDP	–	–	Extension	Extension	
FDL	Flexion	Flexion	Flexion	Flexion	
FDSI	–	–	Flexion	–	
FDPI	–	–	Flexion	–	
Note:

‘–’ indicates that the muscle was inferred not to act around that axis in the model; or was not modelled in that regard.

Moment arms for manus digit I were analysed only for Mussaurus. Regarding muscles crossing the MCP joint (Fig. 10), the EDP and EDS exhibited similar patterns (due to their common paths) of reduced extensor moment arms with MCP joint extension (Fig. 10A), whereas the FDL, Flexor digitorum profundus digiti I (FDPI) and Flexor digitorum brevis superficialis digiti I (FDSI) showed a peak (flexor) moment arm at a moderate joint angle (Fig. 10B). EDP and EDS switched from flexor to extensor moment arms at about −35° of extension, increasing their moment arms until they reached an approximate plateau near 0° (Fig. 10A), whereas FDL (and FDP, FDS) acted fully as a flexor (showing a pattern very similar to that for the MCP joint; Fig. 11B vs. Fig. 10B). For the INP joint (Fig. 11), EDP and EDS showed similar patterns, being extensor muscles that increase extensor moment arm with joint extension, whereas FDL is a fully flexor, also increasing its flexot moment with joint extension.

Discussion

Here, first we compare the results of our forelimb joint ROM analysis in Mussaurus and Crocodylus considering these in light of conclusions from previous studies of this topic in other sauropodomorphs and theropods. Second, we compare the patterns of muscle moment arms in our two study taxa in the context of the evolution of muscle function across Archosauria, comparing with previous qualitative studies dealing with archosaur forelimb myology in which muscle function has been inferred (Meers, 2003; Jasinoski, Russell & Currie, 2006; Langer, Franca & Gabriel, 2007; Allen et al., 2014; Burch, 2014; Klinkhamer et al., 2017). As an important component of this comparison of muscle moment arms, we discuss the effects of altering: (1) joint posture (e.g. reference vs. resting pose); and (2) articular cartilage extent around the elbow in Mussaurus. Finally, we review the evolution of manus pronation in Sauropodomorpha in the light of our results for joint ROM and, where potentially relevant, muscle moment arms.

Joint ROM analysis: implications for the evolution of forelimb posture in sauropodomorphs

Our analysis considered how forelimb joint ROM in Mussaurus differed between the reference and resting poses as well as how the estimated ROM compared with Crocodylus and various saurischian dinosaurs (as previous studies estimated). Here we also evaluate how our findings might reflect potential evolutionary trajectories of maximal forelimb joint ROM in Archosauria, as well as the limitations of these ROM assessments and comparisons. Where relevant, in tandem we also consider our results for muscle moment arms.

The most conservative DOF around the glenohumeral joint in Crocodylus and Mussaurus was pronation and supination (Table 2; Table S3), showing grossly similar values in Crocodylus and Mussaurus. This similarity can partly be attributed to the relatively conserved morphology of the glenohumeral joint surfaces in both taxa, in which the scapular and coracoid lips form an inverted ‘V’ surface. Additionally, the potential ROMs in pronation and supination were relatively independent of the orientation of the glenoid (or pectoral girdle and forelimb) and hence, the same values were estimated for the reference and resting poses of Mussaurus (−25°/25° in pronation/supination), with almost the same values found for Crocodylus johnstoni (−20°/20°). It is reassuring that the latter values were crudely similar to the actual ROM used in vivo during walking in Alligator mississippiensis (−17.9°/27.2° pronation/supination, Baier & Gatesy, 2013). However, our ROM results are smaller than those obtained by Pierce, Clack & Hutchinson (2012; −75°/70°, pronation/supination) for fleshed specimens of Crocodylus niloticus as well as results (approximated as 3D) for Alligator mississippiensis (Hutson & Hutson, 2013). However, skeletonized specimens (e.g. the crocodile used in the present study) might underestimate ROMs vs. fleshed ones (Hutson & Hutson, 2012, 2013), although this is controversial (Pierce, Clack & Hutchinson, 2012; Arnold, Fischer & Nyakatura, 2014; Kambic, Roberts & Gatesy, 2017), probably depending strongly on methods and investigators as well as definitions of 3D joint axes and DOFs. Thus any corroboration of our ROM results for pronation/supination in Crocodylus remains tentative.

In contrast, the remaining glenohumeral DOFs (extension/flexion and abduction/adduction) exhibited different ROM values in both taxa but also in the reference and the resting poses for Mussaurus, which are linked directly to the orientation of the glenohumeral joint. Interestingly, in both the reference and resting poses, Mussaurus displayed a larger ROM for glenohumeral abduction than Crocodylus; whereas Crocodylus had greater capacity for adduction but only if starting in the reference pose (Table 2; Table S3). This difference was probably because of the smooth, broader glenohumeral surface in Mussaurus. It remains problematic that the extent and shape of glenohumeral articular cartilage in Mussaurus is unknown, and we used admittedly simple models of the joint, but our models are provided with this study so that others might build upon our efforts.

A major focus in studies of basal sauropodomorph locomotion is the likelihood of habitual quadrupedalism (Jaekel, 1910; Fraas, 1913; Galton, 1990; Bonnan & Senter, 2007; Bonnan & Yates, 2007; Mallison, 2010a, 2010b; Yates et al., 2010; VanBuren & Bonnan, 2013; Hutson, 2015). However, the ROMs of the forelimb joints depend on the morphology of the articular surfaces (e.g. a wider glenoid surface should allow larger ROMs), and the orientation of such articular surfaces will ultimately determine the way in which ROMs will influence forelimb function (Jenkins, 1993; Gatesy & Baier, 2005). Our ROM analysis of Mussaurus indicated that if the scapula were oriented in an anatomical position of about 55–60° from the horizontal (i.e. caudodorsally), the maximal humeral protraction (i.e. glenohumeral joint abduction) allowed would not pass vertical, which concurs with previous reports for other basal sauropodomorphs (Bonnan & Senter, 2007; Mallison, 2010a, 2010b) and theropods (Senter & Robins, 2005). This inference partially contradicts the possibility of quadrupedalism as a habitual mode of locomotion in early sauropodomorphs such as Mussaurus. However, if the elbow was habitually flexed during locomotion, then quadrupedalism might be more achievable, perhaps with shorter stride lengths (Maidment & Barrett, 2012). However, it remains questionable how flexed the limbs of such a large animal as an adult Mussaurus would have been (Biewener, 1989). Our inference contrasts with the condition inferred in sauropods, in which the glenoid was more ventrally oriented but a caudodorsal scapular blade orientation was maintained (Schwarz, Frey & Meyer, 2007). A ventrally oriented glenoid seems to have allowed sauropods to protract their humerus cranial to vertical, facilitating glenohumeral abduction (i.e. joint movement parallel to the long axis of the glenoid) during quadrupedal locomotion.

Additionally, for the glenohumeral extension/flexion axis, Mussaurus exhibited a combination of peak moment arms at full extension (BB, CBD, SC, TBC, TM and SHP) but also at full flexion (DC, DS, SBS, CBV and TBS), although Crocodylus displayed mostly peak moment arms at full flexion (Fig. 6). In the case of the glenohumeral abduction/adduction axis, Mussaurus had peak moment arms with a more abducted humerus than in Crocodylus (Fig. 7). These results indicated that Mussaurus had greater leverage with a more abducted glenohumeral joint (i.e. humerus) than in Crocodylus, but the consequences of this leverage, and of glenohumeral posture in extension/flexion, are ambiguous although they hint at important functional differences in the forelimb muscles of these two taxa.

Furthermore, although not surprising, the lowest leverages for elbow extensor muscles in both Crocodylus and Mussaurus were at full extension (i.e. a more columnar forelimb). Nonetheless, it is interesting to note that peak extensor moment arms were reached at different elbow joint angles in both taxa. In Crocodylus, peak moment arms were at joint angles of ∼45–55° (Fig. 8A), which implies that leverage could be maximized at a moderately flexed elbow joint. In Mussaurus, elbow extensor moment arms at maximal extension in the resting pose fell drastically to minimal values. Regardless, peak moment arms around the elbow were present at about 90° (Fig. 8A), meaning that maximal muscle leverage was achieved at an even more flexed elbow than in Crocodylus. This mechanical benefit of increased elbow flexion in Mussaurus could be speculated to argue against a forelimb with strong specialization for supportive or locomotor functions (Biewener, 1989), and thus inconsistent with habitual quadrupedalism in Mussaurus.

Overall, Crocodylus and Mussaurus showed interesting postural dependencies of their muscles’ moment arms, but the consequences for quadrupedalism in either taxon are unclear. Likewise, divergent results for moment arm analyses in the hindlimbs of Tyrannosaurus (peak extensor moment arms near full joint extension; Hutchinson et al., 2005) and ostriches (extensor moment arms seeming to be suboptimal for antigravity support in walking and running; Hutchinson et al., 2015) make it difficult, at present, to use these data to test inferences about habitual joint posture. Regardless, all of these studies’ findings reveal how sensitive the moment arms of muscles are to joint orientation. Hence, assuming a constant moment arm is far more risky than checking for this sensitivity.

Similar to pronation and supination around the glenohumeral joint, we found the ROM of flexion and extension around the elbow to be rather conservative between Crocodylus and Mussaurus, even though epiphyseal cartilage in the early sauropodomorph cannot be assessed with great confidence. Full elbow extension (0°) was only allowed (indeed, required) in the reference pose (Table S3), whereas maximal extension in the resting pose was 15–20° for both taxa, avoiding full extension of the elbow. These limits on elbow extension are similar to those found for the basal tetrapod Ichthyostega (Pierce, Clack & Hutchinson, 2012), the crocodylian Alligator (Hutson & Hutson, 2012; Baier & Gatesy, 2013), basal saurischians (Sereno, 1993), quadrupedal ornithischians (Maidment & Barrett, 2012), basal sauropodomorphs (Bonnan & Senter, 2007; Mallison, 2010b; Vargas-Peixoto, Da-Rosa & Franca, 2015), non-avian theropods (Senter & Robins, 2005; White et al., 2015) and birds (Baier, Gatesy & Dial, 2013). Thus our models reject the inference that Mussaurus would have routinely used a fully columnar forelimb pose. This inference also supports the conclusion that no matter if manipulation is being done with fleshed (Hutson & Hutson, 2012) or skeletonized material (Sereno, 1993; Senter & Robins, 2005; Bonnan & Senter, 2007; Mallison, 2010b; Pierce, Clack & Hutchinson, 2012; Vargas-Peixoto, Da-Rosa & Franca, 2015; White et al., 2015), elbow hyperextension close to 180° leads to a high risk of disarticulation. Similarly, maximal elbow flexion was 110–130° in the two taxa (regardless of resting or reference pose), so the total ROM was less in the resting pose vs. the reference pose.

Although we focused on extension/flexion as the major DOF considered for our analysis of elbow and wrist biomechanics, it may be that Mussaurus (like Crocodylus) was capable of other elbow and wrist motions, to some smaller degree. We found some elbow ROM in pronation/supination (∼30°) and abduction/adduction (∼10°) for both taxa (Table 2; Table S3). These findings stand in contrast to some other studies which have inferred negligible ROM for these joints in sauropodomorphs, archosaurs or tetrapods more generally (Senter, 2005; Senter & Robins, 2005; Bonnan & Senter, 2007; Hutson & Hutson, 2012; White et al., 2015). However, the best representations of such joint motions we are aware of are three-dimensional, in vivo quantitative measurements in taxa such as Alligator (Baier & Gatesy, 2013), which found, for example, detectable ROMs of 20–30° for elbow pronation/supination and abduction/adduction during walking. Roughly similar magnitudes of in vivo elbow motion were recently reported for skinks by Nyakatura et al. (2014: their table 2).

Thus there is need for more integration of precise in vivo studies of actual joint motions used by functioning animals (see also Ren et al., 2008: their table 3; Arnold, Fischer & Nyakatura, 2014; Kambic, Roberts & Gatesy, 2017), estimates of maximal ROMs from morphology and other evidence such as fossil trackways (Norman, 1980, 1986; Carrano & Biewener, 1999; Blob & Biewener, 2001; Bates et al., 2008; Arnold, Fischer & Nyakatura, 2014). How much of their arthrologically apparent joint ROMs do real animals use in different behaviours and do they ever naturally use motions that might seem to be ‘dislocations’ based upon osteology or even whole-cadaver studies (Kambic, Roberts & Gatesy, 2017)? How can such data (including improved reconstructions of articular cartilage; see section below) enhance estimates of joint ROM and locomotor function/evolution in extinct tetrapods? At present, we are not convinced that the forelimb joints distal to the glenohumeral joint in archosaurs such as Mussaurus were restricted to purely extension/flexion movements or that other motions were negligible. Yet this controversy is more likely one of a matter of degree, not binary presence/absence of non-parasagittal motions. There is apparently no remaining controversy that extension/flexion motions of the limb joints of dinosaurs and many other archosaurs were the largest ROMs used in vivo or allowed by the joints; our results continue to uphold that inference (Table 2; Table S3).

However, wrist osteology in Mussaurus makes reconstruction of abduction/adduction and pronation/supination ROM difficult to do with much confidence. Indeed, understanding of the mobility of the wrist joints among early sauropodomorphs is limited because there is a lack of information regarding the osteology of the proximal carpus, and the distal carpus is represented by two or three elements with a ‘block’ configuration (Senter & Robins, 2005; Bonnan & Senter, 2007; Mallison, 2010a, 2010b). The only wrist DOF inferred with confidence in Mussaurus is flexion and extension, although even this is speculative. In the case of Crocodylus, the presence of proximal carpal elements allows more confident interpretations regarding pronation/supination and abduction/adduction (Hutson & Hutson, 2014), exhibiting greater ROMs than estimated for Mussaurus (Table 2). Nonetheless, in addition to the issues of cartilage non-preservation noted above, considerable pronation and supination movements have been reported for the hindlimb bones of Alligator mississippiensis during walking (Gatesy, 1991; Blob & Biewener, 2001).

Within digit I in Mussaurus, both the MCP and the INP joints present in early sauropodomorphs (and in dinosaurs in general) displayed ginglymoid, well-defined articular surfaces, contrasting with the flatter (and sometimes pitted) ends of their proximal long bones, for which thick articular cartilage has been inferred (Schwarz, Frey & Meyer, 2007; Holliday et al., 2010; Bonnan et al., 2013). In Mussaurus, we inferred that minimal hyperextension ROM (−25° to −40°) was possible for both joints, with flexion predominating (50–70° maximum), which would be similar to the condition reported for the ungual of digit I of Massospondylus (Cooper, 1981) and the digits of Plateosaurus (Reiss & Mallison, 2014; also White et al., 2015 for the theropod Australovenator). Despite that basal sauropodomorphs share similar manus morphology, more work is needed to test if there are any detectable differences in ROM within this lineage. A limited amount of phalangeal hyperextension has been proposed to be evidence against quadrupedalism because it might also limit the stride length of the forelimb, particularly for the short forelimbs of early sauropodomorphs (Reiss & Mallison, 2014). However, to the degree that any such limitation imposed on stride length existed, it would have been modest relative to the influences of the ROMs of more proximal joints, considering their associated segments’ greater lengths and thus the arcs swept for a given amount of joint ROM. Furthermore, ROMs of the same joints in quadrupedal sauropod(omorph)s still deserve careful study for comparison, as it is questionable whether phalangeal joint motion was important early in the evolution of their quadrupedalism, given the rapid appearance of a columnar, bundled manus in sauropods (Bonnan, 2003; Bonnan & Yates, 2007). Regardless, the mobility of digit I in the manus would be important for other non-locomotor behaviours such as grasping and thus deserves study in more taxa and ultimately in a phylogenetic context.

Archosaur forelimb muscle actions: major differences between sprawling quadrupedalism and erect bipedalism

Although the hindlimbs are/were a terrestrial locomotor module in essentially all archosaurs (living and extinct), the biological role of the forelimbs varies, depending on the locomotor pattern of the organism. Facultative bipedal vertebrates tend to devote the forelimbs to biological roles other than solely body support or locomotion; e.g. manipulation, digging, display and combat. Consequently, among our most interesting findings are estimates of how the mechanical actions of some (but not all) muscles appear to differ between the more sprawling forelimb posture of Crocodylus; presumably at least somewhat similar to the ancestral locomotor pattern of basal archosaurs; to the more derived, erect, at least facultatively bipedal pattern in Mussaurus. More studies are certainly needed to test how much our assumption that Crocodylus’s joint ROM and moment arm patterns are similar to those of ancestral archosaurs (but see Parrish, 1986) and if Mussaurus’s patterns are typical for Sauropodomorpha, especially close to the origin of Sauropoda, but our estimates are important first steps in this direction. Although quantitative functional shifts have been proposed previously for hindlimb muscles in various archosaurs (Hutchinson et al., 2008, 2015; Bates & Schachner, 2012; Maidment & Barrett, 2012), quantitative data for such shifts in the forelimbs of extinct archosaurs have not been reported before.

It is important to note that our study considers muscle actions around the three main DOFs (i.e. pronation/supination, extension/flexion, abduction/adduction) for the glenohumeral joint, which had substantial mobility in both taxa modelled. Past studies, however, have tended to focus on major muscle actions around a single axis, sometimes implicitly assuming that actions around other axes were negligible or unimportant, but more often simplifying muscles to only have one major action (see also Hutchinson et al. (2015) and Rankin, Rubenson & Hutchinson (2016) for similar points regarding 3D actions and broader biomechanical functions—e.g. strut, motor, spring, brake, damper—in the pelvic limb muscles of ostriches). An advantage of our musculoskeletal modelling approach is that, once constructed, actions in any directions can be quantified, and these models could be used in the future to test broader issues about biomechanical functions, biological roles and (with the addition of more musculoskeletal models) comparative evolutionary patterns.

In the above context, the first part of the following section focuses on the influence of the reference vs. resting pose on muscle function, exploring how muscles respond to the shifting from ancestral to derived limb postures in our studied taxa. We then examine how moment arms in the resting pose differ between Crocodylus and Mussaurus, in all cases trying to identify the muscles with actions most influenced by morphology and/or posture. Finally, we reflect on our findings in light of the challenge presented in finding an ‘ideal’ metric by which to normalize moment arm values for comparisons between taxa.

Influence of the reference vs. resting pose on muscle actions

When we compared the muscle actions estimated for the reference and resting poses for our two study taxa, noteworthy differences appeared for the glenohumeral joint, whereas the elbow and wrist joints did not exhibit pronounced differences between the poses (Table 4; Tables S5–S8). This is discussed more in the Online Supplementary Text. Most muscle groups crossing the glenohumeral joint that we analysed in the reference pose had evidence for differences of action between Crocodylus and Mussaurus (7 out of 13 of the muscles in at least one DOF). Furthermore, eight out of 18 muscles acting around the elbow had differences of extensor/flexor action between the two taxa, but no muscle crossing the wrist displayed differences of action (Fig. 9; Fig. S7; Table 4; Table S8). In the case of the resting pose, differences in muscle action between Crocodylus and Mussaurus were slightly less marked for the shoulder, with five out of 13 of the muscles analysed having differences of action in at least one degree of freedom (Table 3; Table S8).

Considering that both musculoskeletal models were set in the same reference pose, in equally sprawled limb orientations, it might seem that the most relevant factor influencing disparity of muscle moment arms is skeletal morphology. While, perhaps unsurprisingly, this appears to generally be correct, limb posture (i.e. behavioural choice of joint orientations) also influences muscle action (Hutchinson et al., 2005, 2015). In particular, we found notable differences in muscle moment arms for glenohumeral extension/flexion between the reference pose and resting pose for our Mussaurus model, with the SHP, TBS, TBC and BB switching actions; becoming mixed or purely extensor/flexor in each case (Table 3; Table S8). These differences in moment arm values between the reference and the resting pose in a single taxon can be explained in terms of posture and ROM. For example, the SHP displayed a mixed action in the reference pose of Mussaurus, remaining an extensor across some of the glenohumeral joint’s ROM, but switching to a flexor at about −40° of extension from the reference pose (i.e. more flexed joint angles). However, the resting pose of Mussaurus had a more restricted ROM (35° vs. 80° in the reference pose) that prevented the SHP from changing into a shoulder flexor (cf. Fig. 6A vs. Fig. S5C). The above examples show how deep the influence of an assumed reference pose could be on the action of a single muscle, particularly for an organism in which such a pose is not anatomically likely, emphasising the need for comparisons made in the context of a biologically plausible (‘resting’) posture, as analysed below.

Functional differences in the resting pose

We focus here on muscles whose actions differed between Crocodylus and Mussaurus in the resting pose. That pose represents a more realistic limb configuration for Mussaurus, allowing us to speculate on underlying causes of such functional changes (e.g. morphology, posture). Furthermore, at the end of this section we compare with data on human forelimb muscle actions, for reasons explained there.

It is important to distinguish between muscles that change their action owing to a shift in their paths (in any posture) because of anatomical changes, and muscles that change their actions because of reorientations of the joints that alter muscle paths. Muscles DC and DS provide a good example of this distinction. These two muscles did not change their paths appreciably from the reference to the resting pose in each taxon, and thus maintained their moment arm patterns, even when moment arms were plotted against a different DOF, to which they are related (Figs. S8 and S9). However, the evolution of dinosaurs involved a counterclockwise (as seen from a right lateral view) rotation of the glenohumeral articular face, as previously noted (Jenkins, 1993; Gatesy & Baier, 2005). This reorientation of the glenoid transformed the functions (i.e. elevation/depression; protraction/retraction) of muscles such as the DC and DS (Figs. 4D–4G and 12A).

For example, in Crocodylus, a flexor action could be incurred by muscles to move the humerus perpendicular to the long axis of the glenoid surface (in the same plane as the vertebral column; in a craniocaudal arc), whereas an adductor action would move the humerus parallel to the glenoid surface (and perpendicular to the vertebral column; in a dorsoventral arc). In a dinosaur such as Mussaurus, the long axis of the glenohumeral joint is not perpendicular to the vertebral column (as in Crocodylus), but caudoventrally (or obliquely) oriented. In birds such glenohumeral reorientation is taken to an extreme, with the long axis more parallel to the vertebral column (Fig. 12B). This reorientation of the shoulder joint along the dinosaurian lineage means that a flexor movement in a crocodile (or other non-dinosaurian archosaur) would protract the humerus. In a typical dinosaur, conversely, a homologous movement (i.e. a movement perpendicular to the long axis of the glenohumeral joint) would elevate the humerus, and this transformation of flexion (for example) would apply to Mussaurus (Fig. 12).

Figure 12 Terminology for homologous joint movements as the glenohumeral articular surface transformed across Archosauria.

(A) Crocodylus and (B) a generalized bird showing homologous joint movement (in this case perpendicular to the long axis of the glenoid) along the extremes of locomotor patterns within Archosauria. (C) Evolution of muscle action around the flexion/extension axis along the ornithodiran line from the ancestral archosaurian pattern for a homologous movement. Same colour/tone indicates the same glenohumeral joint orientation. Line drawings modified from: Crocodylus and generalized bird (Gatesy & Baier, 2005); Pterosauria (Witton, 2015); Ornithischia (Maidment & Barrett, 2011); Camarasaurus (Wilson & Sereno, 1998) and Tawa (Burch, 2014). ad., adduction; fl., flexion.

In the resting pose, SHP was identified as a humeral supinator in Crocodylus but a pronator in Mussaurus, although its origin and insertion sites in both taxa are placed in topologically similar areas on the scapula and humerus, respectively. A role in glenohumeral long axis rotation has not been previously reported for this muscle, to our knowledge. The cause of this divergence in muscle actions would thus be the morphological disparity of the humerus between the crocodile and Mussaurus. The humerus of Crocodylus has a narrow proximal shape in comparison with its shaft, whereas Mussaurus presents an expanded humeral proximal end, as is typical for all early sauropodomorphs (Galton & Upchurch, 2004). Hence the more laterally positioned insertion of the SHP in Mussaurus (Fig. 3) resulted in a sustained pronator action. Conversely, in Crocodylus, SHP’s insertion slightly medial to the humeral midline (Meers, 2003) resulted in a supinator action.

Other morphology-based differences between our study taxa are clearly caused by osteological correlates indicating soft tissue attachments, rather than by general bony geometry. The CBV retained the same shoulder pronator and adductor actions in both taxa (see also Meers, 2003; Jasinoski, Russell & Currie, 2006), but differed strikingly around the extension/flexion axis, being a extensor in Crocodylus and a flexor in Mussaurus (Fig. 6C; Table 3). The protractor and adductor actions of CBV seem to be ancestral for Crocodylia and presumably Archosauria (Meers, 2003; Jasinoski, Russell & Currie, 2006; Burch, 2014), but the derived action in Mussaurus appears to have been incurred by a shift of its path to lie more cranially on the coracoid (Fig. 2).

Some muscles had different action(s) in at least one degree of freedom in the resting pose for Crocodylus and Mussaurus that could best be explained by joint ROMs inferred from our models. TBS was a glenohumeral extensor in Crocodylus, as is ancestral for Archosauria (Meers, 2003; Jasinoski, Russell & Currie, 2006; Burch, 2014). However, TBS had a flexor action in Mussaurus when the joint was moved beyond −60° of extension. As the ROM for glenohumeral maximal extension was limited to −60° in Crocodylus vs. −70° in Mussaurus, this 10° difference in ROM was sufficient to alter the TBS from being a pure extensor to having a mixed action in the latter taxon (Fig. 6C). This pattern of a restricted extension/flexion ROM preventing certain muscles in Mussaurus from switching actions appears common (Fig. 6; Figs. S8 and S9) and would be sensitive to the accuracy of our ROM estimates.

Not all forelimb muscles in our analysis, however, showed different actions in Crocodylus and Mussaurus. Such muscles are interesting, too, because they might have had a conservative function across (at least non-avian) Archosauria. For example, DC and DS were the only muscles acting around the glenohumeral joint that preserved the same action in the three DOFs for both Crocodylus and Mussaurus (Table 3), combining glenohumeral supination, flexion and abduction. These qualitatively identical muscle actions (regardless of their functions in the resting pose; Fig. 12) are reflected by the conservative attachment sites of both DS and DC on the lateral scapular blade and proximal humerus in these two taxa and, more generally, in Archosauria (Meers, 2003; Jasinoski, Russell & Currie, 2006; Langer, Franca & Gabriel, 2007; Remes, 2008; Suzuki & Hayashi, 2010; Burch, 2014; Klinkhamer et al., 2017). The conservative action of the deltoid muscle heads is partially congruent with previous studies that qualitatively inferred crocodile forelimb functions (Meers, 2003; Jasinoski, Russell & Currie, 2006; Allen et al., 2014; Klinkhamer et al., 2017). The TM and TBC, likewise, preserved common shoulder supinator, extensor and abductor actions; whereas the SBS remained a pronator, extensor and adductor; and the SC kept its action as a supinator, flexor and adductor.

Muscles crossing well cranial (i.e. BB, HR and BR) or caudal (i.e. triceps group) to the elbow joint also showed unambiguous actions in Crocodylus and Mussaurus, and more generally in Archosauria (Meers, 2003; Jasinoski, Russell & Currie, 2006; Burch, 2014, 2017; Klinkhamer et al., 2017). The PT muscle was estimated to act as an elbow extensor in Crocodylus but an elbow flexor in Mussaurus (Table 3; Fig. 8C). The more caudal position of the PT’s origin in the former taxon vs. more cranial in the latter (Fig. 2; Table 1) explains this difference. Relative differences in the magnitudes of moment arms, such as the larger normalized values for some muscles (particularly the EDL) acting about the carpus in Crocodylus (Fig. 9), surely relate to the paths of those muscles around the joints and thus to soft tissue morphology and osteological influences on it.

However, less straightforward actions were evident for muscles that originated at either side of the humeral distal condyles, which could experience posture-dependent switches of their actions (Fig. 8). We found complex actions like these for SU, FU, AR, EDL, ECR and ECU, which were elbow extensors in Crocodylus but had mixed actions in Mussaurus (see Meers, 2003; Burch, 2014, 2017 for different interpretations), or FDL, which had a mixed action in Crocodylus but was purely a flexor in Mussaurus. The differences in ROM values between both taxa seemingly did not affect the actions of these muscles (Table 3; Table S8). Instead, the actions of these muscles were extremely sensitive to placements of their origin sites (Fig. 2; Fig. S1). The main problem resulting from this sensitivity is the uncertainty about the location of the origin sites in Mussaurus, which are obscured by pitting and other artefacts left by the articular cartilage. Below, we consider the effects of missing cartilage on our general conclusions about forelimb biomechanics in Mussaurus.

Of all forelimb muscles in tetrapods, the actions in humans are the best understood, particularly using these musculoskeletal modelling frameworks and often in conjunction with validation methods such as magnetic resonance imaging or ‘tendon travel’ experiments (see references below for an introduction). Although a detailed comparison with Mussaurus (or Crocodylus) is far beyond the scope of this study, general patterns of similarities and differences in muscle actions are evident. First, however, the actions must be considered in light of joint ROMs. Compared with Mussaurus and Crocodylus (Table 2), humans (data from Murray, Delp & Buchanan, 1995; Murray, Buchanan & Delp, 2000; Holzbaur, Murray & Delp, 2005; Rankin & Neptune, 2012) have much larger total ROMs for extension/flexion (∼120°) and long-axis rotation (∼105°) of the shoulder (let alone the highly mobile scapula). Humans also certainly have a greater range of forearm pronation/supination (∼160° in models although far less is used in vivo; closer to 90°). Otherwise, the ROMs of the human shoulder in ab/adduction (∼65°) and elbow and wrist in flexion/extension (∼100° and 90° respectively; also ∼35° of wrist ab/adduction or ‘deviation’) are roughly similar to the archosaurs we studied.

Keeping these similarities and differences in joint ROMs in mind, and focusing mainly on extensor/flexor action, there are interesting similarities. In all three taxa, the deltoid muscles tend to increase their shoulder flexor moment arms with increasing joint flexion, and TM muscles tend to have shoulder extensor moment arms that peak at moderate angles of shoulder flexion (Ackland et al., 2008; vs. our Fig. 6). Yet differences in shoulder muscle actions also stand out. For instance, human subscapularis (SBS) muscles have high flexor moment arms in extended shoulder positions, moving towards extensor action as the shoulder becomes strongly flexed (Ackland et al., 2008), whereas the SBS muscles in our archosaur models are predominantly extensors (Fig. 6).

Around the elbow joint, human and archosaur triceps muscles have more flattened extensor moment arm vs. joint angle curves, whereas the biceps and brachialis elbow flexors have pronounced peaks for their elbow flexor moment arms at moderate flexion angles (Murray, Buchanan & Delp, 2000; our Fig. 8). Data from humans hint that these consistent patterns may relate to different usages of the active force-length curves of these muscles: elbow flexors tend to range far in their relative lengths across the ascending limb and plateau of that curve, whereas triceps muscles tend to remain closer to the curve’s plateau (Murray, Buchanan & Delp, 2000: their Fig. 5). In other words, the former muscles may be more specialized for length change whereas the triceps muscles, even in bipeds (such as Mussaurus?), may remain specialized for antigravity functions such as high isometric force production. This speculation deserves testing with more direct measurements in living archosaurs as well as modelling investigations; moment arms alone are only tantalizing in this regard. However, other muscles acting around the elbow show some divergent patterns: for example, the PT is a weak elbow flexor in humans but has a moderate extensor or more mixed action in our archosaur models (see above), and the very transformed M. brachioradialis in humans is an elbow flexor with a large moment arm (and capacity for length change), unlike the corresponding ECR+ECU muscles in our models which are estimated to be elbow extensors in Crocodylus and mixed in Mussaurus (Murray, Delp & Buchanan, 1995, their Fig. 4; vs. our Fig. 8).

Finally, we see few clear similarities for muscle actions around the wrist joints of the three taxa. The wrist joint of humans has long flexor muscles (Mm. flexores carpi radialis et ulnaris) comparable with our models’ FDL in terms of anatomy. Their moment arms for flexion increase slightly with wrist flexion angle—but in our archosaur models, these muscles exhibit weaker flexion moment arms with increasing flexion and even switch to extensors at extremely flexed joint angles (Fig. 9B; this latter finding may be due to implausible via points and/or joint axes in our models, though). A similar pattern holds for wrist extensors: human data indicate stronger extensor moment arms in extended poses, unlike our models’ moment arms, which peak at moderate joint angles close to the reference pose, and may switch to flexor action in Crocodylus if the wrist is strongly extended (González, Buchanan & Delp, 1997, their Fig. 4; vs. our Fig. 9A).

These patterns in human, Crocodylus and Mussaurus forelimb muscles broadly match results from some other animals, especially quadrupedal mammals such as hares and greyhounds (Williams, Wilson & Payne, 2007; Williams et al., 2008), reinforcing that some similarities may be generalizations for tetrapods or amniotes whereas others may be specializations particular to mammals or archosaurs (or bipeds). Unfortunately too few data exist, especially for non-mammals, to test these ideas and more focus is needed on collecting new data from experiments and models of various taxa, joints and muscles. We have only made a small step here towards integrating these disparate sources of data to understand the evolution of muscle actions and functions.

Implications of normalization metrics used for moment arm comparisons

The general patterns that we present here for moment arm postural changes and muscle actions (e.g. extensor/flexor/mixed) in Crocodylus and Mussaurus are not influenced by our choice of segment length as a normalizing metric. However, comparisons of the absolute and normalized values of moment arms are influenced by the vast differences in morphology and posture (and phylogenetic divergence times) between our two study taxa. Considering that relative rather than absolute values were most needed here, and the latter problems of moment arm comparisons across disparate taxa, we have generally not emphasized those values of moment arms. As Table S2 shows, the ratios of corresponding segment lengths from our two models vary little; between ∼1.9 and 2 (Mussaurus has relatively longer proximal segments, especially humerus). These ratios slightly complicate proximodistal comparisons across the limbs, which were not a focus of our study. More problematically, the ideal normalization metric would be body mass (to remove size influences), but this is unknown for Mussaurus. Using the minimal humeral and femoral circumferences of our Mussaurus specimen and equation 2 from Campione & Evans (2012), we obtained an estimate of 1,486 kg body mass, 73.6 times that of our Crocodylus specimen, or 4.13 times if the cube root of body mass were desired as an approximately linear normalizer (reducing moment arms in metres to dimensionless units as in this study’s main results). The equation used for body mass estimation has ‘error bars’ of appreciable size, but our focus was more on qualitative comparisons of muscle actions than quantitative ones (especially absolute magnitudes except where exceptional). Additionally, we provide our models, results and normalizing metrics here, so it is feasible for future studies to inspect the effects of this assumption in more detail if desired. We do not expect that our conclusions would be considerably altered by using a different normalizing metric. If more taxa were included in our analysis, this issue would become more important to consider, so we raise it here but do not elaborate further. An alternative approach would be to present ratios of moment arms (e.g. extensor/flexor vs. abductor/adductor) but for simplicity we did not add this analysis. Studies of human moment arms have suggested that the ideal normalization metrics may even vary with the relative distance of the origin or insertion from the joint centre, and thus bone lengths, diameters or circumferences might thus be insufficient for normalization (Murray, Buchanan & Delp, 2002).

Sensitivity analysis: influence of cartilage volume on moment arms at the elbow joint

One of the major challenges inherent to soft tissue reconstructions in extinct archosaurs is the reliable inference of sites of origin and insertion of muscles, as well as 3D paths between them. However, some muscles leave notable scars and protuberances on the bone surfaces, and thus inferences about their existence and locations become less speculative than those muscles for which no osteological correlates exist (Bryant & Seymour, 1990; Witmer, 1995). This is relevant not only for the correct interpretation of the anatomy of the animal, but also has a profound impact on biomechanical inferences based on the musculoskeletal anatomy (Hutchinson et al., 2005). The moment arms of some limb muscles are very sensitive to the inferences made about muscle attachments and paths, especially insertions. Fortunately, in many cases the more concentrated nature of insertions (vs. more diffuse nature of proximal origins of muscles, which tend to taper towards their distal insertions) means that the insertions have clearer scars and thus locations, thereby sometimes reducing concerns about that sensitivity. Yet as noted above for the elbow (e.g. distal humerus), missing articular/epiphyseal cartilages is one clear case where there is cause for special concern and thus attention to potential sensitivity of the moment arms of muscles that cross the elbow joint.

The inference of epiphyseal cartilage in extinct archosaur limbs has been the subject of debate and speculation since the late 1800s (Owen, 1875; Cope, 1878; Osborn, 1898), with the main focus of discussions centred on the estimated volume occupied by the original cartilage cap (Holliday et al., 2010; Bonnan et al., 2013; Reiss & Mallison, 2014; White et al., 2015). However, inferences about the impact of cartilage volume in functional studies have received less attention (but see Gatesy, Bäker & Hutchinson, 2009; Fujiwara, Taru & Suzuki, 2010; Tsai & Holliday, 2014; Taylor & Wedel, 2013). The lost cartilage during the process of fossilization in dinosaurs is evident in the simplified epiphyseal surfaces of the long bones. These missing surfaces also complicate interpretations of musculoskeletal biomechanics because they affect the assumed length of the segment(s) analysed and the shape of the articular facet(s) as well (Holliday et al., 2010; Bonnan et al., 2013). Similarly, functional analyses dealing with joint articulations of limb bones in archosaurs have mostly focused on how the absence (or presence) of cartilage can influence the ranges of motion (ROM) of joints (Mallison, 2010b; Hutson & Hutson, 2012, 2013, 2014; Reiss & Mallison, 2014), although some studies opted for a bone-on-bone analysis, arguing that speculation about cartilage extent was simply excessive (White et al., 2015). Overall, there is virtually no information on how unpreserved cartilage volumes may affect muscle function in archosaur limbs. To address this matter in our musculoskeletal model by testing how the estimated moment arms were influenced by articular cartilage morphology, we varied the effective cartilage volume of the epiphyses by altering the wrapping surfaces of muscles crossing the elbow joint.

Increasing or decreasing the radius of the cylinder that muscles traversing the distal humeral condyles must wrap around when they contacted it represented an increase/decrease of the epiphyseal cartilage assumed for the elbow joint. Subsequently, the radius of this wrapping cylinder was then increased or decreased by 25% of its original value (Fig. 13; Table S11), and in each case we recalculated all of the affected muscles’ moment arms (Table S10). These changes required some adjustments of muscle points to prevent muscle paths from penetrating the wrapping cylinder (Figs. 13A and 13C).

Figure 13 Sensitivity analysis of elbow extensor muscles in Mussaurus patagonicus (right elbow in lateral view; resting pose).

Cartilage diameter is shown for (A) reduced by 25% from original, (B) original and (C) enlarged by 25% from original.

Our results from this sensitivity analysis showed that altering the effective cartilage volume at the elbow did not affect the qualitative pattern of moment arms for extensor musculature (Fig. 14). The triceps muscle moment arms showed the same human-like pattern described above (increase of extensor moment arm with flexion past the resting pose, then decrease past ∼90°). In spite of the similar trajectories of the moment arm curves, altering the volume of the hypothetical cartilage cap did (unsurprisingly) alter moment arm values around the elbow. Reduction of the wrapping surface at the elbow by 25% of its radius considerably decreased the overall moment arms of extensor muscles. At full elbow extension, moment arms with 25% reduction of the wrapping surface (i.e. ‘cartilage’) were smaller by about 0.01 m from the original values, and almost 0.02 m smaller than the model with enlarged (+25%) wrapping surfaces (Table S10). At full flexion, however, leverage differences increased, involving larger values for the model with increased wrapping surface, as expected (Fig. 14; Table S10). These results were expected because muscle moment arms can be defined as the minimal distance between the line of action of a muscle-tendon complex and the centre of rotation of a joint (An et al., 1984; Pandy, 1999). On average, moment arms calculated for the 25% reduced ‘cartilage cap’ experienced a decrease of 15% of moment arms relative to the non-altered model’s mean moment arm (Table S10), whereas an increase of 25% to the ‘cartilage cap’ resulted in a 14% increase of moment arm values. Hence, reduction of the cartilage cap (as a wrapping surface) around a given joint should lead to a reduction (although potentially non-proportional) of the moment arms for that joint. Thus, although consideration of cartilage volume and estimated shape in ROM analysis deserves scrutiny because of potential for subjectivity (White et al., 2015), our results demonstrate that missing articular cartilage will influence muscle moment arm variations, highlighting the importance of epiphyseal caps for inferences about muscle functions and evolution.

Figure 14 Sensitivity analysis of elbow extensor muscles in Mussaurus patagonicus.

Moment arms around the elbow joint (not normalized), plotted against extension/flexion joint angles for Mussaurus in the resting pose; for various elbow cartilage assumptions (see Fig. 13 and Materials and Methods). Negative moment arms correspond to extension. Zero elbow angle corresponds to full extension, while larger angles correspond to flexion. For muscle abbreviations see Table 1.

Manus pronation in Mussaurus and the evolution of quadrupedalism in Sauropodomorpha

The biped–quadruped transition in Sauropodomorpha was linked with the dramatic postural changes that evolved from the smaller sauropodomorph ancestors to the gigantic sauropods (Upchurch, Barrett & Galton, 2007). Such changes also involved a series of anatomical transformations, including increased body mass and a forward shift of the body’s centre of mass (Bates et al., 2016), modification of limb proportions (Wilson, 2002), and successive addition of sacral vertebrae (Wilson & Sereno, 1998; Pol, Garrido & Cerda, 2011), among others. Furthermore, over the past 15 years, pronation of the manus has been proposed as a critical anatomical feature associated with the acquisition of quadrupedal locomotion in different lineages of Dinosauria (Bonnan, 2003; Bonnan & Senter, 2007; Bonnan & Yates, 2007; Maidment & Barrett, 2012; VanBuren & Bonnan, 2013; Hutson, 2015). The growing consensus is that increased manus pronation originated during the early evolution of large-bodied, quadrupedal and graviportal sauropodomorphs for improved support against gravity. This consensus exists in contrast to the evolution of pronation capabilities in extant taxa, including some lizards and mammals, in which pronation seems to have been correlated with increased arboreality at small body sizes (Matthew, 1904; Haines, 1958; VanBuren & Bonnan, 2013; Hutson, 2015; Hutson & Hutson, 2017).

Bonnan (2003) and Bonnan & Yates (2007) hypothesized that at least a semi-pronated manus in sauropodomorph dinosaurs was facilitated, among others, by the evolution of the craniolateral process of the proximal ulna that accommodated the radius in a cranial (not medial) position relative to the ulna proximally, and in a medial position relative to the ulna distally. Then, the evolution of the characteristic U-shaped manus in eusauropods may have originated in relation to increased pronation (Bonnan & Yates, 2007, p. 166). In contrast, Hutson (2015) and Hutson & Hutson (2015) proposed that the evolution of the craniolateral process of the ulna was a specialization to immobilize the proximal radioulnar joint. Either way, increased pronation of the manus should have aided the forelimbs to generate craniocaudally directed propulsive or braking forces that roughly paralleled the actions of the pes in a parasagittal plane (Bonnan, 2003). Excluding (putatively ancestrally) bipedal forms such as Panphagia and Saturnalia, most non-sauropod sauropodomorphs are hypothesized to have been either facultative quadrupeds or bipeds, although few studies have delved deeply into this topic.

Moreover, although widely cited in the literature, the terms ‘active’ and ‘passive’ pronation have been scarcely defined, existing mostly in an implicit fashion (see Bonnan & Senter, 2007; Bonnan & Yates, 2007; VanBuren & Bonnan, 2013). Hutson & Hutson (2015, 2017), however, defined active forearm pronation as anteromedial long-axis rotation. Here we define active pronation as the muscle-driven ability to rotate the manus around its longitudinal axis, from pronation to supination, by any kind of rearrangements of the antebrachial bones. Active pronation may facilitate facultative quadrupedalism. Passive pronation implies a manus fixed into pronation, with no clear ability to supinate, leading to obligate quadrupedalism.

Bonnan & Senter (2007) suggested that the early Jurassic massopodans Plateosaurus and Massospondyus had poor abilities for quadrupedal locomotion (thus favouring bipedalism) based on the restricted ROMs of their limb joints and the morphology of their radius and ulna (i.e. straight radius, not crossing the ulna), which may have precluded active or passive pronation. Additionally, ROM analysis performed on a 3D skeleton of Plateosaurus showed that radius rotation around the ulna was impossible, mainly because of its oval-shaped (not circular) proximal end, precluding pronational capabilities and thus quadrupedal locomotion (Mallison, 2010a, 2010b). Nonetheless, a permanently semi-pronated manus was not ruled out for Plateosaurus (Mallison, 2010b). In contrast to this, active pronation has been reported in some therian clades with an oval radial epiphysis (Hutson & Hutson, 2017), implying that rotation of the radius against the ulna should be analysed considering not just one parameter, such as the shape of the proximal radius (see also VanBuren & Bonnan, 2013), but also a broader range of traits.

In contrast, a permanently semi-pronated manus is inferred to have been present in Melanorosaurus (Bonnan & Yates, 2007), a sauropodomorph closely related to Sauropoda (Yates, 2007; Pol, Garrido & Cerda, 2011; Otero et al., 2015). In the latter studies, a semi-pronated manus was concluded to have evolved at least in sauropodomorphs close to Sauropoda, at the base of the ‘quadrupedal clade.’ This clade retained other clearly ‘prosauropod-like’ forelimb features (e.g. an arched metacarpus, three manus claws, and a medially divergent pollex), indicating a potential decoupling of manus shape and quadrupedalism. Other features hint at a functional connection between forelimb morphology (e.g. presence of a craniolateral process on the ulna) and manus shape (i.e. presence of an arched, rather than bundled, metacarpus) (Bonnan & Yates, 2007). Regardless, how the forelimbs of early sauropodomorphs were used for functions other than purely locomotion has hitherto not been convincingly addressed, and the functional steps that ultimately produced the derived locomotor mechanisms present in Sauropoda remain obscure, deserving testing with a wider sample of taxa.

The forelimb of Mussaurus patagonicus is particularly interesting because it displays a combination of plesiomorphic and derived features. For example, it has sauropodomorph plesiomorphies such as expanded humeral epiphyses, a metacarpus that is arranged into a gentle arch, and a robust metacarpal I with a medially divergent pollex (Otero & Pol, 2013). Contrastingly, the evolution of an incipient craniolateral process of the proximal ulna (indicating a rearrangement of the radius relative to the ulna; Bonnan & Yates, 2007) is a derived feature in Mussaurus, shared with other sauropodiforms (e.g. Aardonyx, Sefapanosaurus, Melanorosaurus) and sauropods. Moreover, Mussaurus is phylogenetically placed at the base of the sauropodiform clade (Otero & Pol, 2013; McPhee et al., 2015; Otero et al., 2015), constituting an intermediate taxon to test pronation capabilities between the plesiomorphic pattern present in non-sauropodiform sauropodomorphs (i.e. Massospondylus, Plateosaurus) and the derived pattern inferred for the closest relatives of Sauropoda (i.e. Melanorosaurus).

To estimate the potential for manus pronation in Mussaurus, we used our 3D musculoskeletal model to evaluate how the radius might have been accommodated against the ulna and which antebrachial configurations Mussaurus could have adopted in order to achieve some amounts of manus pronation. Recent studies demonstrated that the morphology of the radius is an important determinant of pronation capabilities, such as the presence of radial shaft curvature (allowing the radius to cross the ulna) and a rounded proximal articular face (permitting the radius to rotate around the proximal end of the ulna during active pronation); a condition typical of extant mammals (VanBuren & Bonnan, 2013; Hutson & Hutson, 2017). Nonetheless, the presence of a mediolaterally expanded radial head and the absence of radial shaft curvature may have precluded active manus pronation in most dinosaurs (VanBuren & Bonnan, 2013) (and perhaps other archosaurs; Hutson, 2015). Moreover, another feature would have prevented active manus pronation specifically in sauropodomorph dinosaurs. The distal end of the radius of several sauropodomorphs across the transition to Sauropoda had a prominent tubercle on the caudodistal surface, which was suggested to be an osteological correlate of the radioulnar ligament’s attachment (Remes, 2008; Yates et al., 2010; Otero & Pol, 2013; McPhee et al., 2014; Otero et al., 2015). This caudodistal tubercle of the radius is a feature characteristic of basal sauropodiforms such as Mussaurus, Aardonyx, Sefapanosaurus, Melanorosaurus and Antetonitrus (Fig. 15; McPhee et al., 2014; Otero et al., 2015), and it is also present in the basal sauropod Tazoudasaurus (Allain & Aquesbi, 2008: Fig. 22).

Figure 15 Antebrachial bones of Mussaurus patagonicus.

Radius and ulna (A) showing the articulation of the distal ends in medial (B), distomedial (C), dorsolateral (D) and lateral (E) views. Not to scale.

Digital manipulation of our 3D model of Mussaurus in our ROM analyses showed that there was limited possibility of radial movement against the ulna both proximally and distally. The elliptical proximal surface of the radius precluded pronation/supination and the distal tubercle would have locked the distal radius and ulna, placing the former cranial to the latter. Furthermore, the radius of Mussaurus is rather straight, making radial crossing around the ulna impossible. Considering these constraints, the most likely way to articulate the radius and ulna in an anatomically plausible way was with the radius cranial to the ulna proximally, and slightly medially distally, as previously suggested by Bonnan (2003). Nonetheless, with this antebrachial configuration, we infer that appreciable manus pronation (via radioulnar rotation) was not possible in Mussaurus. Thus, the only way to achieve some degree of pronation in Mussaurus was through pronation (internal/medial rotation) of the whole antebrachium as a single unit (i.e. around the elbow joint) by up to about −30° (Table 2). With this configuration, some degree of manus pronation might have been achievable, although far from the full pronation of the manus that might be consistent with permanently quadrupedal locomotion (Fig. 16). This does not mean that Mussaurus actually did perform pronation in this way, but it is a possibility, considering that the articular surfaces of the distal humerus and proximal radius seem to allow it (albeit cartilage shape is unknown). Considering that crocodylians appear able to conduct some similar long-axis rotation (Baier & Gatesy, 2013), this is not an outrageous proposition.

Figure 16 Antebrachial articulations of Mussaurus patagonicus.

Non-pronated (i.e. semi-pronated, sensu Hutson & Hutson, 2015, 2017) (A, C, E) and semi-pronated (i.e. 30° of medial rotation/pronation from A) (B, D, F) poses depicting the relationships among antebrachial bones. Radius and ulna in proximal (A, B) views, forelimb in cranial (C, D) views, and manus in proximal (E, F) views.

All of the above features in our ROM and morphological analysis of Mussaurus support the inference that mobility of the radius against the ulna was severely restricted in most non-sauropodiform sauropodomorph dinosaurs, making active pronation of the manus through antebrachial rotation highly unlikely. Nonetheless, our ROM analysis showed that active semi-pronation might have been possible in Mussaurus through pronation of the whole antebrachium at the elbow (−30°). This potential pronation ability constitutes a novel, but tentative, finding for basal sauropodomorphs, consistent with the inference that facultative quadrupedalism should not be ruled out for this taxon (and perhaps close relatives), although obligate quadrupedalism was unlikely.

However, active pronation would presumably have been muscle-driven; and thus moment arms would be important for driving such motions or controlling postures. Our moment arm analysis for pronation/supination around the elbow (Fig. S10) was interesting in two ways in this regard. First, we found that only the FU muscle could pronate the elbow joint (increasing its pronator moment arm as the elbow became less pronated); all other antebrachial muscles were consistently supinators. Second, almost all supinator muscles acting around the elbow (except the EDL, ECR and ECU) had maximal supinator moment arms at −30° pronation; reducing with increased elbow supination. It is not clear what pronation/supination moments the elbows of Mussaurus would have needed to support if used in an antigravity role during quadrupedal locomotion, but our model indicates a potentially overall greater leverage of the forelimb (for actions of pronation or supination) in a non-pronated orientation; i.e. closer to 0° (Table S12). Note that the elbows of Mussaurus in an antigravity role almost certainly would have had to sustain extensor muscle actions (e.g. triceps muscle group activity) and the activations of these muscles would have created supinator moments around the elbow that the FU muscle alone might have had difficulty opposing to maintain a semi-pronated posture (i.e. when its moment arms were minimal but its antagonists’ were maximal). Together, this evidence does not strongly favour the ability of Mussaurus to actively pronate its elbow joints, but it remains a possibility based on our ROM results in particular. As we cautioned above, however, these inferences are strongly contingent on our assumptions and conclusions about elbow joint morphology and articular cartilage in Mussaurus.

The evolution of a pronated manus has been postulated to have begun at least prior to the rise of sauropods, at the origin of the quadrupedal sauropodiform clade (i.e. Melanorosaurus, Bonnan & Yates, 2007; Yates et al., 2010). Aardonyx, a basal member relative to Melanorosaurus outside the quadrupedal clade, was proposed to have had some features that preceded quadrupedal locomotion in sauropodomorphs, such as an incipient craniolateral process of the ulna and a rather straight femoral shaft (Yates et al., 2010), but the question of how much earlier this evolution began has remained unresolved. We conclude, considering past studies as well as our new data for Mussaurus, that full, passive manus pronation was not present at the base of Sauropodiformes (sensu Sereno, 2007), but instead much closer to the origin of Sauropoda than previously thought (see also Yates et al., 2010). However, we cannot exclude some capacity for active pronation in Mussaurus and presumably some other sauropodiforms, as a potential intermediate state in this transformational series of forelimb function. One alternative would be that quadrupedalism did not merely evolve once in the sauropodomorph lineage, but rather that mosaic evolution in early Sauropodiformes resulted in some taxa tending to use quadrupedalism more often than others did. Ultimately, reconstruction of the origin, and perhaps stepwise acquisition, of manus pronation in Sauropodomorpha will depend upon further analyses using not only qualitative, descriptive approaches but also quantitative, explicitly three-dimensional methods such as the one adopted here.

Conclusion

We have presented the first quantitative evaluation of forelimb muscle actions in a sauropodomorph dinosaur, and combined this with assessments of joint mobility and phylogenetic inferences. Comparisons made with Crocodylus, which represents a mode of locomotion that is closer to the presumed ancestral state for Archosauria, frame our study in a broader context to better understand major locomotor shifts in the sauropodomorph line within Archosauria, including a review of the major literature.

Analysis of moment arms revealed that, first: major differences of muscle actions between Crocodylus and Mussaurus are evident at the glenohumeral joint, and such changes are correlated with the morphology of the scapula and the orientation of the glenohumeral articulation in both taxa (supporting the inference that many of these changes occurred from Archosauria to Dinosauria/Sauropodomorpha). Second, forelimb posture has great impact on moment arm values, more so in many cases than morphology. Third, our analysis of reference vs. resting pose in the studied taxa demonstrated how extensive the influence of such poses could be on the action of a single muscle, particularly for an organism in which that pose is not anatomically likely (such as Mussaurus), requiring the need for comparisons made in a context of biologically plausible posture (i.e. resting pose). Fourth, caution is warranted when comparing organisms with shifted joint coordinate systems, like Crocodylus (sprawled limb/vertical scapula) and Mussaurus (erect limb/caudoventrally inclined scapula), in which the same homologous movement, like extension/flexion (i.e. the humerus moving perpendicular to the long axis of the glenoid), actually corresponds to protraction/retraction (i.e. the humerus moving in a cranial/caudal direction relative to the ground) in the former and elevation/depression in the latter (i.e. the humerus moving dorsal/ventral relative to the ground, see Fig. 4), as raised by Gatesy & Baier (2005). Fifth, sensitivity analysis conducted on Mussaurus’ elbow joint confirms that more extensive cartilage volume would increase the moment arms of elbow extensor muscles, in particular.

Finally, habitual quadrupedalism in Mussaurus is not supported by our joint ROM analysis, in which glenohumeral protraction was found to be severely restricted. Additionally, some small amount of active pronation of the manus might have been possible in Mussaurus, and perhaps in other earlier sauropodomorphs, via long-axis rotation at the elbow to achieve semi-pronation of the whole antebrachium (not rotation of the radius around the ulna, as previously thought). In summary, then, the rise of quadrupedalism in Sauropoda would be linked not only to manus pronation, which should have occurred very close to the node Sauropoda. This quadrupedalism was also enabled by shifting forelimb morphology as a whole, allowing larger extension/flexion excursions of the glenohumeral joint and a more columnar forelimb posture. Our open modelling methods allow others to inspect and build upon these findings. Further methodological progress in refining how joint ROM is estimated and its biological implications for certain behaviours, combined with data from how living animals move their joints with muscles (especially involving how muscle areas and lengths contribute to joint moments and locomotion), is needed to build consensus in this field, particularly regarding the evolution of manus pronation vs. quadrupedalism in Archosauria.

Supplemental Information

Supplemental Information 1 Table S1. Wrapping surfaces used for the musculoskeletal models of Crocodylus johnstoni and Mussaurus patagonicus.

Click here for additional data file.

Supplemental Information 2 Table S2. Assessment of normalizing metrics used in moment arm analyses.

‘MC II’ is metacarpal II length. Lengths are in metres (m); for Mussaurus the sum circumference in mm was used to estimate body mass (see Discussion), and the ratio of ‘linearized’ body masses to the 0.33exponent is shown in the final column’s final entry (for comparison to normalizing ratios of segment lengths in the ‘ratio’ row to the left).

Click here for additional data file.

Supplemental Information 3 Table S3. Ranges of motion (ROMs) of forelimb in Mussaurus and Crocodylus about each degree of freedom analyzed in this study in the reference pose.

Click here for additional data file.

Supplemental Information 4 Table S4. Results for glenohumeral joint moment arms (in metres) of major muscle groups in the resting pose for Mussaurus and Crocodylus.

Click here for additional data file.

Supplemental Information 5 Table S5. Results for elbow joint moment arms (in metres) of major muscle groups in the resting pose for Mussaurus and Crocodylus.

Click here for additional data file.

Supplemental Information 6 Table S6. Results for metacarpo-phalangeal and inter-phalangeal joint moment arms (in metres) of major muscle groups for digit one of Mussaurus.

Click here for additional data file.

Supplemental Information 7 Table S7. Results for elbow and wrist joint moment arms (in metres) of major muscle groups in the reference pose for Mussaurus and Crocodylus.

Click here for additional data file.

Supplemental Information 8 Table S8. Muscle actions for Crocodylus and Mussaurus in the reference pose.

Click here for additional data file.

Supplemental Information 9 Table S9. Results for shoulder joint moment arms (in metres) of major muscle groups in the reference pose for Mussaurus and Crocodylus.

Click here for additional data file.

Supplemental Information 10 Table S10. Results of the sensitivity analysis for elbow joint moment arms (in metres) of the major extensor muscle group in the resting pose in Mussaurus.

Click here for additional data file.

Supplemental Information 11 Table S11. Wrapping surfaces used for the musculoskeletal model of Mussaurus patagonicus in the sensitivity analysis.

For additional muscle abbreviations and details see Table 1 and Table S1’s caption.

Click here for additional data file.

Supplemental Information 12 Table S12. Results for elbow joint moment arms (in metres) of major muscle groups in the resting pose in Mussaurus in a pronated position (by 30°) for long axis rotation.

Click here for additional data file.

Supplemental Information 13 Figure S1. Three-dimensional musculoskeletal model of the right forelimb of Mussaurus patagonicus (elbow area; medial view) showing via points.

Click here for additional data file.

Supplemental Information 14 Figure S2. Isolated wrapping objects used in this study. Cylinder (A), ellipsoid (B) and torus (C), in multiple views.

Click here for additional data file.

Supplemental Information 15 Figure S3. Three-dimensional models and wrapping surfaces.

Three-dimensional musculoskeletal models of the right forelimbs of Mussaurus patagonicus (A–C) and Crocodylus johnstoni (D–F) in the resting pose, showing wrapping objects used in this study in lateral (A, D), medial (B, E) and caudomedial (C, F) views. Scale bar: 10 cm. Compare with Fig. 3.

Click here for additional data file.

Supplemental Information 16 Figure S4. Pronation/supination moment arms around the glenohumeral joint, normalized to humerus segment length, plotted against pronation/supination joint angles for Crocodylus and Mussaurus in the reference pose.

(A) mostly pronator; (B) mostly supinators; (C) mixed pronators/supinators. Negative moment arms and glenohumeral angles correspond to pronation, while positive values correspond to supination. For muscle abbreviations see Table 1.

Click here for additional data file.

Supplemental Information 17 Figure S5. Extension/flexion moment arms around the glenohumeral joint, normalized to humerus segment length, plotted against extension/flexion joint angles for Crocodylus and Mussaurus in the reference pose.

(A) mostly extensors; (B) mostly flexors; (C) mixed extensors/flexors. Negative moment arms and glenohumeral angles correspond to extension, while positive values correspond to flexion. For muscle abbreviations see Table 1.

Click here for additional data file.

Supplemental Information 18 Figure S6. Abduction/adduction moment arms around the glenohumeral joint, normalized to humerus segment length, plotted against abduction/adduction joint angles for Crocodylus and Mussaurus in the reference pose.

(A) mostly abductors; (B) mostly adductors; (C) mixed abductors/adductors. Negative moment arms and glenohumeral angles correspond to abduction, while positive values correspond to adduction. For muscle abbreviations see Table 1.

Click here for additional data file.

Supplemental Information 19 Figure S7. Extension/flexion moment arms around the elbow joint, normalized to radius-ulna segment length, plotted against extension/flexion joint angles for Crocodylus and Mussaurus in the reference pose.

(A) extensors; (B) flexors; (C) mixed extensors/flexors. Negative moment arms correspond to extension, while positive values correspond to flexion. Zero elbow angle corresponds to full extension, while larger angles correspond to flexion. For muscle abbreviations see Table 1.

Click here for additional data file.

Supplemental Information 20 Figure S8. Pronation/supination moment arms around the glenohumeral joint, normalized to humerus segment length, plotted against extension/flexion joint angles for Crocodylus and Mussaurus in the resting pose.

(A) mostly pronators; (B) mostly supinators; (C) mixed pronators/supinators. Negative moment arms and glenohumeral angles correspond to pronation, while positive values correspond to supination. For muscle abbreviations see Table 1.

Click here for additional data file.

Supplemental Information 21 Figure S9. Abduction/adduction moment arms around the glenohumeral joint, normalized to humerus segment length, plotted against flexion/extension joint angles for Crocodylus and Mussaurus in the resting pose.

(A) mostly abductors; (B) mostly adductors; (C) mixed abductors/adductors. Negative moment arms and glenohumeral angles correspond to abduction, while positive values correspond to adduction. For muscle abbreviations see Table 1.

Click here for additional data file.

Supplemental Information 22 Figure S10. Pronation/supination moment arms around the elbow joint (not normalized), plotted against pronation/supination joint angles for Mussaurus in the resting pose.

(A) mostly pronators; (B) mostly supinators; (C) mixed pronators/supinators. Negative moment arms and joint angles correspond to pronation, whilst positive values correspond to supination. Zero elbow angle corresponds to a neutral position in between pronation/supination. For muscle abbreviations see Table 1.

Click here for additional data file.

Supplemental Information 23 Online Supplementary text.

Click here for additional data file.

N. Muñoz (Museo de La Plata) digitized the forelimb of Mussaurus. Three-dimensional models of Mussaurus were possible thanks to the freely available software Meshlab (3D-CoForm project) and 3D Studio (Autodesk Education Community). We thank Kent Vliet and David Kledzik for providing the crocodile specimen for study and Larry Witmer for the use of his laboratory and access to a CT scanner for imaging that specimen. Reviews by Susannah Maidment and Joel Hutson greatly improved the manuscript.

Additional Information and Declarations

Competing Interests

Author Contributions

Data Availability

John Hutchinson is an Academic Editor for PeerJ.

Alejandro Otero conceived and designed the experiments, performed the experiments, analysed the data, contributed reagents/materials/analysis tools, wrote the paper, prepared figures and/or tables, reviewed drafts of the paper.

Vivian Allen performed the experiments, analysed the data, contributed reagents/materials/analysis tools, prepared figures and/or tables, reviewed drafts of the paper.

Diego Pol analysed the data, contributed reagents/materials/analysis tools, reviewed drafts of the paper.

John R. Hutchinson conceived and designed the experiments, performed the experiments, analysed the data, contributed reagents/materials/analysis tools, wrote the paper, prepared figures and/or tables, reviewed drafts of the paper.

The following information was supplied regarding data availability:

Otero, Alejandro; Allen, Vivian; Pol, Diego; Hutchinson, John (2017): Crocodylus musculoskeletal models. figshare.

https://doi.org/10.6084/m9.figshare.4928696.v1

Otero, Alejandro; Allen, Vivian; Pol, Diego; Hutchinson, John (2017): Mussaurus musculoskeletal models. figshare.

https://doi.org/10.6084/m9.figshare.4928684.v1

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
