# Peer review of "Forelimb muscle and joint actions in Archosauria: insights from Crocodylus johnstoni (Pseudosuchia) and Mussaurus patagonicus (Sauropodomorpha)"

_PeerJ, doi:10.7717/peerj.3976_

## Round 0.1 · original submission · Minor Revisions

· Academic Editor

Minor Revisions

Dear authors,

Both reviewers are highly positive about your study, noting that the research aims are clear, that the manuscript has been carefully prepared, and that the authors have presented a careful and considered discussion of the results and their implications. They suggest that this study would make an important contribution to the literature following minor revisions, and I agree with their comments. Please pay particular attention in your response to the comments regarding interpretations relating to pronation (see Reviewer #2), and quadrupedality (Reviewer #1). Reviewer #1 has also made some helpful suggestions to improve the figures.

I look forward to receiving your revised manuscript.

·

Basic reporting

This is a well-written and well-executed paper examining muscle moment arms and joint ranges of motion in the forelimb muscles of an extant ‘sprawling’ archosaur and an extinct ‘parasagittal’ archosaur.

Basic reporting

The English is generally clear and unambiguous throughout. There is some confusion of UK vs. US English throughout; e.g. meters (US), but modelling and centre (UK). I have identified a few typos in the text and picked them on the attached PDF. The paper is set in an appropriate context, the literature is appropriately referenced, and the structure is appropriate.

The figures containing moment arm graphs are far too small. I was unable to read the text in the figures, and while they might be reproduced slightly larger in final print, they are still way too small. I suggest breaking the moment arm figures up and making the text at least 10 points at final size.

The manuscript states in one place that models are available upon request, but in another that they are included with the study. My preference would be to see the models included with the study, but that is the prerogative of the authors.

You state at the beginning that extension is negative, but throughout there is confusion about flexion and extension and which is negative.

Maybe I missed something but I can’t see any figure captions for the supplementary information figures.

Experimental design

The research question is clearly framed, well defined, relevant and meaningful. It has been performed to a high standard and the methods have been described in sufficient detail.

Validity of the findings

The data appears robust, is clearly described, and is appropriately interpreted.

Conclusions drawn are linked to the original research questions and supported by the results.

I have some comments regarding your interpretations about quadrupedality.

1) You state that not being able to flex the humerus anterior of vertical would preclude quadrupedalism. I don’t think that’s the case. In obligate quadrupedal ornithischians, I don’t think the humerus could have been protracted past vertical because of the location of the humeral head on the caudal surface of the humerus and the morphology of the glenoid. Because the elbow is strongly flexed, however, this doesn’t prevent quadrupedalism. I’ve said this somewhere in print and can probably dig it out for you. But these animals were clearly quadrupedal. It must have meant they had a very short stride, admittedly, but I don’t think it precludes quadrupedality;

2) A pronated manus isn’t necessarily necessary for quadrupedality. Trackways of iguanodontids show quadrupedal locomotion with an apparently supinated manus;

3) In sauropodomorphs, quadrupedality seems to be considered to only have occurred once. I’m not sure we necessarily have to assume this. It’s possible that there was mosaic evolution of the forelimb in sauropodomorph evolution and that facultative quadrupedality could have been present in some forms and not in others. Thus contradictory findings might be expected on the sauropodomorph ‘stem’ when examining different basal sauropodomorphs.

Comments for the author

There are many minor comments on the attached pdf.

·

Basic reporting

1) Clear, unambiguous, professional English language used throughout.
Excellent. Just some minor typographical errors were noted with sticky notes.

2) Intro & background to show context. Literature well referenced & relevant.
The introduction and background are good, but the section on pronation appears to have overlooked the most recently published tests of the references that they rely on most heavily in their results and discussion. I provided detailed comments, explanations, and references to ask for some clarifications in terminology and their interpretations of pronative ability in their study specimens.

3) The structure of the submitted article should conform to one of the templates.
No comments.

4) Figures should be relevant to the content of the article.
The figures are good, and I trust that the published version will indicate which side is Crocodylus, and which is Mussaurus.

5) The submission should be 'self-contained,' should represent an appropriate 'unit of publication,' and should include all results relevant to the hypothesis.
No comments

6) All appropriate raw data has been made available in accordance with our Data Sharing policy.
No comments

Experimental design

1) Original primary research.
- No comments.

2) Research question well defined, relevant & meaningful. It is stated how research fills an identified knowledge gap.
- In regards to their stated knowledge gap for dinosaurian pronation, I inserted comments as needed to help identify what some of the newest studies have tested in relation to their research questions. Hopefully these comments can help identify how their pronation study fits in with the latest research.

3) The investigation must have been conducted rigorously and to a high technical standard.
- Excellent.

4) Methods described with sufficient detail & information to replicate.
- Excellent.

5) The research must have been conducted in conformity with the prevailing ethical standards in the field.
- No comments

Validity of the findings

1) The data should be robust, statistically sound, and controlled.
- No comments

2) The data on which the conclusions are based must be provided or made available in an acceptable discipline-specific repository.
- No comments

3) The conclusions should be appropriately stated, should be connected to the original question investigated, and should be limited to those supported by the results.
- Excellent, but with some inserted concerns as to their support for their pronation results.

4) Speculation is welcomed, but should be identified as such.
- No comments

5) Decisions are not made based on any subjective determination of impact, degree of advance, novelty, being of interest to only a niche audience, etc. Replication experiments are encouraged (provided the rationale for the replication, and how it adds value to the literature, is clearly described); however, we do not allow the ‘pointless’ repetition of well known, widely accepted results.
- No comments.

6) Negative / inconclusive results are acceptable.
- No comments

Comments for the author

Very nice piece of writing - This paper is carefully written, with thoughtful consideration to different points of view and terminologies so common to myological/arthrological studies. The layout is very organized and flows nicely; the figures are easy to interpret. I am impressed with the support for your interpretations of how to categorize movements based off of your moment arms; this is the type of quantitative support needed to help standardize some of the terminology for sprawling versus parasagittal clades. I inserted a number of sticky notes and highlighted text throughout the pdf markup. Most of them are minor suggestions, typographical error notations, and questions. My only concern was with the pronation aspect of the study. My latest research is testing some of your primary references on forearm pronation in dinosaurs, and I noticed that your arguments could benefit by incorporating them. I tried to outline how your interpretations of forearm pronation could be affected by this research.

---

## Round 0.2 · accepted · Accept

· Academic Editor

Accept

Thank you for carefully addressing the comments of both reviewers in this revised version. Your revisions have dealt with the minor comments raised previously and I think your manuscript is now ready to be published.

·

Basic reporting

no comment

Experimental design

no comment

Validity of the findings

no comment

Comments for the author

I believe that the changes in discussion and interpretation are more than satisfactory in helping identify how your study fits in with the latest pronation research.